# RETHINKING ADVERSARIAL ATTACKS AS PROTECTION AGAINST DIFFUSION-BASED MIMICRY

## ABSTRACT

Diffusion models have demonstrated a remarkable capability to edit or imitate images, which has raised concerns regarding the safeguarding of intellectual property. To address these concerns, the adoption of adversarial attacks, which introduce adversarial perturbations that can fool the targeted diffusion model into protected images, has emerged as a viable solution. Consequently, diffusion models, like many other deep network models, are believed to be susceptible to adversarial attacks. However, in this work, we draw attention to an important oversight in existing research, as all previous studies have focused solely on attacking latent diffusion models (LDMs), neglecting adversarial examples for diffusion models in the pixel space diffusion models (PDMs). Through extensive experiments, we demonstrate that nearly all existing adversarial attack methods designed for LDMs, as well as adaptive attacks designed for PDMs, fail when applied to PDMs. We attribute the vulnerability of LDMs to their encoders, indicating that diffusion models exhibit strong robustness against adversarial attacks. Building upon this insight, we find that PDMs can be used as an off-the-shelf purifier to effectively eliminate adversarial patterns generated by LDMs, thereby maintaining the integrity of images. Notably, we highlight that most existing protection methods can be easily bypassed using PDM-based purification. We hope our findings prompt a reevaluation of adversarial samples for diffusion models as potential protection methods.

## 1    INTRODUCTION

Generative diffusion models (DMs) (Ho et al., 2020; Song et al., 2020; Rombach et al., 2022) have achieved great success in generating images with high fidelity. However, this remarkable generative capability of diffusion models is accompanied by safety concerns (Zhang et al., 2023a), especially on the unauthorized editing or imitation of personal images such as portraits or individual artworks (Andersen, 2023; Setty, 2023). Recent works (Liang et al., 2023; Shan et al., 2023; Salman et al., 2023; Xue et al., 2023; Zheng et al., 2023; Chen et al., 2024; Ahn et al., 2024; Liu et al., 2023) show that adversarial samples (adv-samples) for diffusion models can be applied as a protection against malicious editing. Small perturbations generated by conventional methods in adversarial machine learning (Madry et al., 2018; Goodfellow et al., 2014) can effectively fool popular diffusion models such as Stable Diffusion (Rombach et al., 2022) to produce chaotic results when an imitation attempt is made. However, a significantly overlooked aspect is that all the existing works focus on latent diffusion models (LDMs) and the pixel-space diffusion models (PDMs) are not studied. For LDMs, perturbations are not directly introduced to the input of the diffusion models. Instead, they are applied externally and propagated through an encoder. It has been shown that the encoder and decoder of LDMs are vulnerable to adversarial perturbations (Zhang et al., 2023b; Xue et al., 2023), which means that the adv-samples for LDMs have a very different mechanism of action compared to the adv-samples for PDMs. Moreover, some existing works (Liang and Wu, 2023; Salman et al., 2023) show that using an encoder-specific loss can enhance the adversarial attack, (Xue et al., 2023) further demonstrating that the encoder is the bottleneck for attacking LDMs. Building upon this observation, in this paper, we draw attention to the problem of rethinking existing adversarial attack methods for diffusion models by asking the question:

*Can we generate adversarial examples for PDMs as we did for LDMs?*

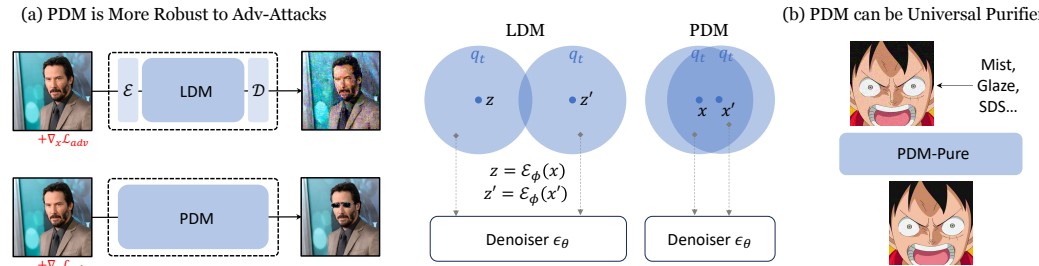

Figure 1: Overview: (a) Recent protection approaches based on adversarial perturbation against latent diffusion models (LDMs) cannot be used in pixel-space diffusion models (PDMs); The underlying reason is that the encoder of the Latent Diffusion Model (LDM) amplifies the perturbations, causing the inputs to the denoiser to have significantly different distributions. In contrast, the inputs of the PDM maintain large overlap, showing robustness. (b) A strong PDM can be used as a universal purifier to effectively remove the protective perturbation generated by existing protection methods. (Best viewed with zoom-in on a computer)

We address this question by systematically investigating adv-samples for PDMs. We conduct experiments on various LDMs or PDMs with different network architectures (e.g. U-Net (Ho et al., 2020), Transformer (Peebles and Xie, 2023)), different training datasets, and different input resolutions (e.g. 64, 256, 512). Through extensive experiments, we demonstrate that all the existing methods we tested (Liang and Wu, 2023; Zheng et al., 2023; Shan et al., 2023; Xue et al., 2023; Chen et al., 2024; Salman et al., 2023; Liang et al., 2023), developed to attack LDMs, fail to generate effective adv-samples for PDMs. Moreover, we conduct adaptive attacks for PDMs, applying strategies like gradient averaging and attacking the intermediate features, but none of the attacks can effectively change the reverse diffusion process the way the do to fool LDMs. This implies that PDMs are more adversarially robust than we think.

Building on the insight that PDMs are strongly robust against adversarial perturbations, we further propose PDM-Pure, a universal purifier that can effectively remove the protective perturbations of different scales (e.g. Mist-v2 (Zheng et al., 2023) and Glaze (Shan et al., 2023)) based on PDMs trained on large datasets. Through extensive experiments, we demonstrate that PDM-Pure achieves way better performance than all baseline methods.

To summarize, the pixel is a barrier to adversarial attack (Figure 1); the diffusion process in the pixel space makes PDMs much more robust than LDMs. This property of PDMs also makes real protection against the misusage of diffusion models difficult since: (1) no existing attacks have proven effective in attacking PDMs, which means no protection can be achieved by fooling a PDM, (2) all the existing protections against LDMs can be easily purified using a strong PDM. Our contributions are listed below.

1. We observe that most existing works on adversarial examples for protection focus on LDMs. Adversarial attacks against PDMs are **largely overlooked** in this field.
2. We fill in the gap in the literature by conducting extensive experiments on various LDMs and PDMs. We discover that all the existing methods **fail** to attack the PDMs, indicating that PDMs are much more adversarially robust than LDMs.
3. Based on this novel insight, we propose a simple yet effective framework termed PDM-Pure that applies strong PDMs as **a universal purifier** to remove attack-agnostic adversarial perturbations, easily bypassing almost all existing protective methods.

## 2 RELATED WORKS

**Adversarial Examples for DMs** Adversarial samples (Goodfellow et al., 2014; Carlini and Wagner, 2017; Shan et al., 2023) are clean data samples perturbed by an imperceptible small noise that can fool deep neural networks into making wrong decisions. Under white-box conditions, gradient-based methods are widely used to generate adv-samples. Among them, the projected gradient descent (PGD) algorithm (Madry et al., 2018) is one of the most effective methods. Recent works (Liang et al., 2023;

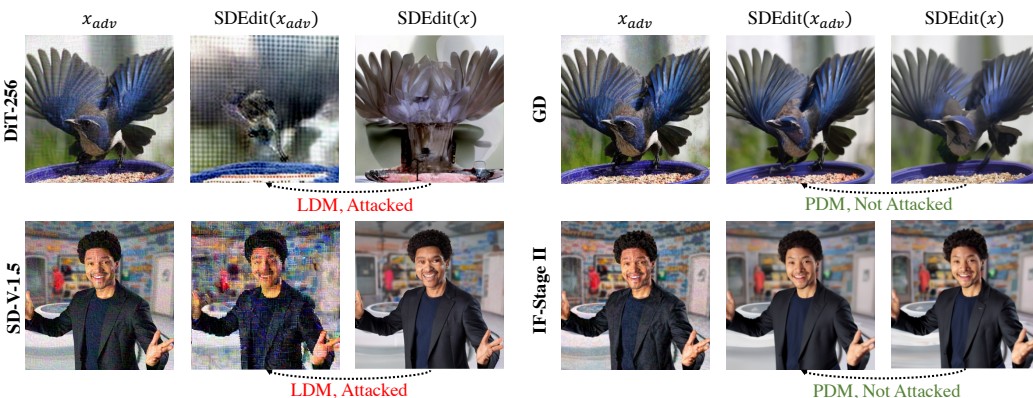

Figure 2: **PDMs Cannot be Attacked as LDMs**: LDMs can be easily fooled by running PGD to fool the denoising loss, but PDMs cannot be easily fooled. DiT (Peebles and Xie, 2023) and SD (Rombach et al., 2022) are LDMs, GD (Dhariwal and Nichol, 2021) AND IF-Stage-II (Shonenkov et al.) are PDMs (Best viewed with zoom-in)

Salman et al., 2023) show that it is also easy to find adv-samples for diffusion models (AdvDM): with a proper loss to attack the denoising process, the perturbed image can fool the diffusion models to generate chaotic images when operating diffusion-based mimicry. Furthermore, many improved algorithms (Zheng et al., 2023; Chen et al., 2024; Xue et al., 2023) have been proposed to generate better AdvDM samples. However, to our best knowledge, all the AdvDM methods listed above are used on LDMs, and those for the PDMs are rarely explored.

**Adversarial Perturbation as Protection**    Adversarial perturbation against DMs turns out to be an effective method to safeguard images against unauthorized editing (Liang et al., 2023; Shan et al., 2023; Salman et al., 2023; Xue et al., 2023; Zheng et al., 2023; Chen et al., 2024; Ahn et al., 2024; Liu et al., 2023). It has found applications (e.g., Glaze (Shan et al., 2023) and Mist (Zheng et al., 2023; Liang and Wu, 2023)) for individual artists to protect their creations. SDS-attack (Xue et al., 2023) further investigates the mechanism behind the attack and proposes some tools to make the protection more effective. However, they are limited to protecting LDMs only. In addition, some works (Zhao et al., 2023; Sandoval-Segura et al., 2023) find that these protective perturbations can be purified. For instance, GrIDPure (Zhao et al., 2023) find that DiffPure (Nie et al., 2022) can be used to purify the adversarial patterns, but they did not realize that the reason behind this is the robustness of PDMs.

## 3    PRELIMINARIES

**Generative Diffusion Models**    The generative diffusion model  (Ho et al., 2020; Song et al., 2020) is one type of generative model, and it has demonstrated remarkable generative capabilities in numerous fields such as images (Rombach et al., 2022; Balaji et al., 2022), 3D data (Poole et al., 2023; Lin et al., 2022), video (Ho et al., 2022; Singer et al., 2022), stories (Pan et al., 2022; Rahman et al., 2023) and music (Mittal et al., 2021; Huang et al., 2023) generation. Diffusion models, like other generative models, are parametrized models $p_\theta(\hat{x}_0)$ that can estimate an unknown distribution $q(x_0)$. For image generation tasks, $q(x_0)$ is the distribution of real images.

There are two processes involved in a diffusion model, a forward diffusion process and a reverse denoising process. The forward diffusion process progressively injects noise into the clean image, and the $t$-th step diffusion is formulated as $q(x_t \mid x_{t-1}) = \mathcal{N}(x_t; \sqrt{1 - \beta_t} x_{t-1}, \beta_t \mathbf{I})$. Accumulating the noise, we have $q_t(x_t \mid x_0) = \mathcal{N}(x_t; \sqrt{\bar{\alpha}_t} x_{t-1}, (1 - \bar{\alpha}_t)\mathbf{I})$. Here $\beta_t$ growing from 0 to 1 are pre-defined values, $\alpha_t = 1 - \beta_t$, and $\bar{\alpha}_t = \Pi_{s=1}^t \alpha_s$. Finally, $x_T$ will become approximately an isotropic Gaussian random variable when $\bar{\alpha}_t \to 0$.

Inversely, $p_\theta(\hat{x}_{t-1}|\hat{x}_t)$ can generate samples from Gaussian $\hat{x}_T \sim \mathcal{N}(0, \mathbf{I})$, where $p_\theta$ is re-parameterized by learning a noise estimator $\epsilon_\theta$, the training loss is $\mathbb{E}_{t,x_0,\epsilon}[\lambda(t)\|\epsilon_\theta(x_t, t) - \epsilon\|^2]$

weighted by $\lambda(t)$, where $\epsilon$ is the noise used to diffuse $x_0$ following $q_t(x_t|x_0)$. Finally, by iteratively applying $p_\theta(\hat{x}_{t-1}|\hat{x}_t)$, we can sample realistic images following $p_\theta(\hat{x}_0)$.

Since the above diffusion process operates directly in the pixel space, we call such diffusion models Pixel-Space Diffusion Models (PDMs). Another popular choice is to move the diffusion process into the latent space to make it more scalable, resulting in the Latent Diffusion Models (LDMs) (Rombach et al., 2022). More specifically, LDMs first use an encoder $\mathcal{E}_\phi$ parameterized by $\phi$ to encode $x_0$ into a latent variable $z_0 = \mathcal{E}_\phi(x_0)$. The denoising diffusion process is the same as PDMs. At the end of the denoising process, $\hat{z}_0$ can be projected back to the pixel space using a decoder $\mathcal{D}_\psi$ parameterized by $\psi$ as $\hat{x}_0 = \mathcal{D}_\psi(\hat{z}_0)$.

**Adversarial Examples for Diffusion Models**  Recent works (Salman et al., 2023; Liang et al., 2023) find that adding small perturbations to clean images will make the diffusion models perform badly in noise prediction, and further generate chaotic results in tasks like image editing and customized generation. The adversarial perturbations for LDMs can be generated by optimizing the Monte-Carlo-based adversarial loss:

$$\mathcal{L}_{adv}(x) = \mathbb{E}_{t,\epsilon}\mathbb{E}_{z_t \sim q_t(\mathcal{E}_\phi(x))}\|\epsilon_\theta(z_t, t) - \epsilon\|_2^2. \tag{1}$$

Other encoder-based losses (Shan et al., 2023; Liang and Wu, 2023; Zheng et al., 2023; Xue et al., 2023) further enhance the attack to make it more effective. With the carefully designed adversarial loss, one can run Projected Gradient Descent (PGD) (Madry et al., 2018) with $\ell_\infty$ budget $\delta$ to generate adversarial perturbations:

$$x^{k+1} = \mathcal{P}_{B_\infty(x^0,\delta)}\left[x^k + \eta\,\mathrm{sign}\nabla_{x^k}\mathcal{L}_{adv}(x^k)\right] \tag{2}$$

In the above equation, $\mathcal{P}_{B_\infty(x^0,\delta)}(\cdot)$ is the projection operator on the $\ell_\infty$ ball, where $x^0$ is the clean image to be perturbed. We use superscript $x^k$ to represent the iterations of the PGD and subscript $x_t$ for the diffusion steps.

## 4 RETHINKING ADVERSARIAL EXAMPLES FOR DIFFUSION MODELS

### 4.1 DIFFUSION MODELS DEMONSTRATE STRONG ADVERSARIAL ROBUSTNESS

While there are many approaches that adopt adversarial perturbation to fool diffusion models, most of them focus only on latent diffusion models due to the wide impact of Stable Diffusion; no attempts have been made to attack PDMs. This lack of investigation may mislead us to conclude that diffusion models, like most deep neural networks, are vulnerable to adversarial perturbations, and that the algorithms used for LDMs can be transferred to PDMs by simply applying the same adversarial loss in the pixel space formulated as: $\mathcal{L}_{adv}(x) = \mathbb{E}_{t,\epsilon}\mathbb{E}_{x_t \sim q_t(x)}\|\epsilon_\theta(x_t, t) - \epsilon\|_2^2$.

However, we show through experiments that PDMs are robust against this form of attack (Figure 2), which means all the existing attacks against diffusion models are, in fact, special cases of attacks against the LDMs only. We conduct extensive experiments on popular LDMs and PDMs structures including Diffusion Transformer (DiT), Guided Diffusion (GD), Stable Diffusion (SD), and Deep-Floyd (IF), and demonstrate in Table 1 that only the LDMs can be attacked and PDMs are not as susceptible to adversarial perturbations: for PDMs, the image quality does not significantly decrease due to the perturbation both visually and quantitatively. More details and analysis can be found in the experiment section.

Prior to this study, there may have been a prevailing belief that diffusion models could be easily deceived. However, our research reveals an important distinction: it is the LDMs that exhibit vulnerability, while the PDMs demonstrate significantly higher adversarial robustness.

| Models | FID-score↑ | | | SSIM ↓ | | | LPIPS ↑ | | | IA-Score ↓ | | | Type |
|---|---|---|---|---|---|---|---|---|---|---|---|---|---|
| $\delta = 4/255$ | Clean | Adv | Δ | Clean | Adv | Δ | Clean | Adv | Δ | Clean | Adv | Δ | |
| DiT-256 | 131 | 167 | +36 | 0.37 | 0.35 | -0.02 | 0.44 | 0.54 | +0.10 | 0.74 | 0.70 | -0.04 | LDM |
| SD-V-1.4 | 44 | 114 | +70 | 0.68 | 0.55 | -0.13 | 0.22 | 0.46 | +0.24 | 0.92 | 0.84 | -0.08 | LDM |
| SD-V-1.5 | 45 | 113 | +68 | 0.73 | 0.59 | -0.14 | 0.20 | 0.38 | +0.138 | 0.94 | 0.89 | -0.05 | LDM |
| GD-ImageNet | 109 | 109 | +0 | 0.66 | 0.66 | -0.00 | 0.21 | 0.21 | +0.00 | 0.90 | 0.90 | -0.00 | PDM |
| IF-I | 186 | 187 | +1 | 0.59 | 0.58 | -0.01 | 0.14 | 0.14 | +0.00 | 0.86 | 0.86 | -0.00 | PDM |
| IF-II | 85 | 87 | +2 | 0.84 | 0.84 | -0.00 | 0.15 | 0.15 | +0.00 | 0.91 | 0.91 | -0.00 | PDM |
| $\delta = 8/255$ | Clean | Adv | Δ | Clean | Adv | Δ | Clean | Adv | Δ | Clean | Adv | Δ | |
| DiT-256 | 131 | 186 | +55 | 0.37 | 0.31 | -0.06 | 0.44 | 0.63 | +0.19 | 0.74 | 0.66 | -0.08 | LDM |
| SD-V-1.4 | 44 | 178 | +134 | 0.68 | 0.44 | -0.24 | 0.22 | 0.60 | +0.38 | 0.92 | 0.78 | -0.14 | LDM |
| SD-V-1.5 | 45 | 179 | +134 | 0.73 | 0.49 | -0.24 | 0.20 | 0.51 | +0.31 | 0.94 | 0.84 | -0.10 | LDM |
| GD-ImageNet | 109 | 110 | +1 | 0.66 | 0.64 | -0.02 | 0.21 | 0.22 | +0.01 | 0.90 | 0.90 | -0.00 | PDM |
| IF-I | 186 | 188 | +2 | 0.59 | 0.59 | -0.00 | 0.14 | 0.14 | +0.00 | 0.86 | 0.86 | +0.00 | PDM |
| IF-II | 85 | 82 | -3 | 0.84 | 0.83 | -0.01 | 0.15 | 0.16 | +0.01 | 0.91 | 0.92 | +0.01 | PDM |
| $\delta = 16/255$ | clean | adv | Δ | clean | adv | Δ | clean | adv | Δ | clean | adv | Δ | |
| DiT-256 | 131 | 220 | +89 | 0.37 | 0.26 | -0.11 | 0.44 | 0.70 | +0.26 | 0.74 | 0.63 | -0.11 | LDM |
| SD-V-1.4 | 44 | 225 | +181 | 0.68 | 0.34 | -0.34 | 0.22 | 0.68 | +0.46 | 0.92 | 0.72 | -0.20 | LDM |
| SD-V-1.5 | 45 | 226 | +181 | 0.73 | 0.37 | -0.36 | 0.20 | 0.62 | +0.42 | 0.94 | 0.78 | -0.16 | LDM |
| GD-ImageNet | 109 | 110 | +1 | 0.66 | 0.57 | -0.09 | 0.21 | 0.26 | +0.05 | 0.90 | 0.89 | -0.01 | PDM |
| IF-I | 186 | 188 | +2 | 0.59 | 0.58 | -0.01 | 0.14 | 0.15 | +0.01 | 0.86 | 0.87 | +0.01 | PDM |
| IF-II | 85 | 86 | +1 | 0.84 | 0.76 | -0.08 | 0.15 | 0.21 | +0.06 | 0.91 | 0.95 | +0.04 | PDM |

Table 1: **Quantitative Measurement of PGD-based Adv-Attacks for LDMs and PDMs**: gradient-based diffusion attacks can attack LDMs effectively, making the difference Δ across all evaluation metrics between edited clean image and edited adversarial image large, which means the quality of edited images drops dramatically. However, the PDMs are not affected much by the crafted adversarial perturbations, showing small Δ before and after the attacks.

## 4.2 ADAPTIVE ATTACKS FOR PIXEL-SPACE DIFFUSION MODELS

To further test the robustness of pixel-space diffusion models, we proceed by designing more adaptive attacks for PDMs. We adopt some design code from (Tramer et al., 2020) to craft adaptive attacks. We first divide the attacks into two categories (C1): attack the full pipeline, which is an end-to-end attack for the targeted editing pipeline. (C2): use diffusion loss as the objective, which follows Equation 1.

Then we try other tricks e.g. applying Expectation over Transformation (EoT) (Athalye et al., 2018), using a targeted attack, and a latent attack (attacking the intermediate layers). We collect the following attacks to test the robustness of Guided Diffusion (GD), including:

- Attack (1) / (2): (C1) with / without EoT
- Attack (3) / (4): (C2) with targeted / untargeted loss without EoT
- Attack (5) / (6): The above two attacks with EoT
- Attack (7) / (8): Latent attack / Latent attack+ in (Shih et al., 2024)

Attacks (1)–(6) are largely ineffective against PDMs, suggesting that end-to-end or Expectation over Transformation (EoT) attacks are unlikely to yield better results. As demonstrated in Figure 3, all crafted perturbations fail to induce chaotic generation outcomes in PDMs.

Recent work by (Shih et al., 2024) introduces latent attacks that can effectively deceive diffusion models. The core idea is to target the intermediate layers of the U-Net architecture in Guided Diffusion (GD). While this type of attack appears capable of misleading the PDM to edit the object as something different (see Figure 4), it suffers from two major limitations: The perturbation magnitude is excessively large, with $\ell_\infty(\delta) > 150/255$. As a result, the appearance of the objects is significantly altered and further degraded by added Gaussian noise. Consequently, the diffusion model will not be able to correctly identify the object. For instance, as shown in the last block of Figure 4, when large Gaussian noise is introduced, the diffusion model mistakenly identifies the chicken as a turtle. Additionally, such latent attacks are ineffective when the editing strength is low, indicating that the attack mechanism heavily relies on the magnitude of noise applied. In contrast, attacks against Latent

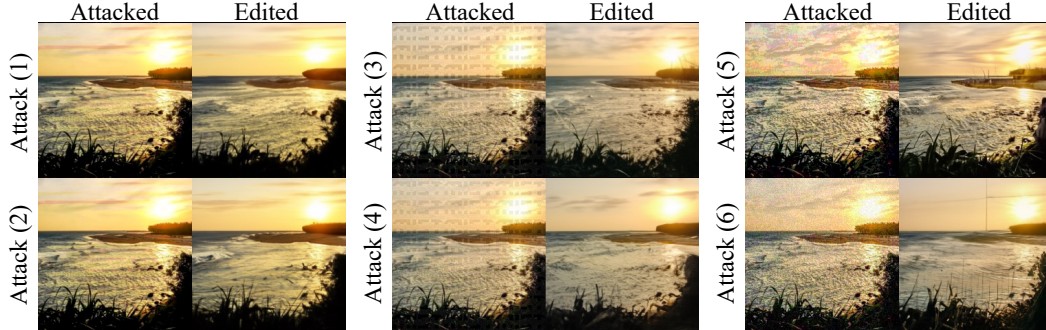

Figure 3: **Crafting Adaptive Attacks for PDMs**: PDM shows robustness against end-to-end attacks and sampling based attacks, for EoT settings. We use the images in (Zheng et al., 2023) as the targeted image in the pixel space.

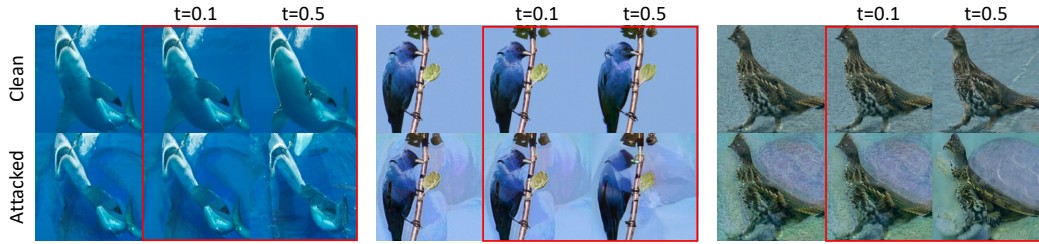

Figure 4: **Latent Attacks for PDMs**: (Shih et al., 2024) proposes to attack the intermediate feature of the denoiser, and use a additional encoder-decoder to regularize the perturbation. This kind of attack need large perturbation $\ell_\infty > 150/255$, and it barely work for small editing steps.

Diffusion Models (LDMs) can remain effective even with small perturbation steps, as they are capable of crafting strong adversarial attacks despite limited noise being added.

### 4.3 LATENT DIFFUSION MODEL IS VULNERABLE BECAUSE OF THE ENCODER

The previous two sections demonstrate that PDMs exhibit significantly stronger empirical robustness compared to LDMs. Rather than providing a theoretical proof of the robustness of the diffusion process in pixel space (which is challenging to establish for DNN-based systems), we offer an intuitive explanation for why PDMs exhibit greater resilience.

The vulnerability of the LDMs is caused by the vulnerability of the latent space (Xue et al., 2023), meaning that although we may set budgets for perturbations in the pixel space, the perturbations in the latent space can be large. In (Xue et al., 2023), the authors show statistics of perturbations in the latent space over the perturbations in the pixel space and this value $\frac{|z-z'|}{|x-x'|}$ can be as large as 10, making the inputs into the denoiser ($z_t = q_t(z), z'_t = q_t(z')$) have smaller overlap (Figure 1 Middle). In contrast, the inputs into PDMs ($x_t = q_t(x), x'_t = q_t(x')$) will still have large overlap, since $x$ and $x'$ are close to each other due to the limited attack budget.

If we decompose the attacks on LDMs into two categories: (a) attacking the encoder and (b) attacking the diffusion model. We observe that the former is due to the encoder's adversarial vulnerability, while the latter results from a significant domain shift. Essentially, the input changes so drastically that it diverges from the distribution of the training environment, leading to reduced performance and robustness.

Almost all the copyright protection perturbations (Shan et al., 2023; Liang and Wu, 2023; Zheng et al., 2023) are based on the insight that it is easy to craft adversarial examples to fool diffusion models. We need to rethink the adversarial samples for diffusion models since there are a lot of PDMs that cannot be attacked easily. Next, we show that PDMs can be utilized to purify all adversarial

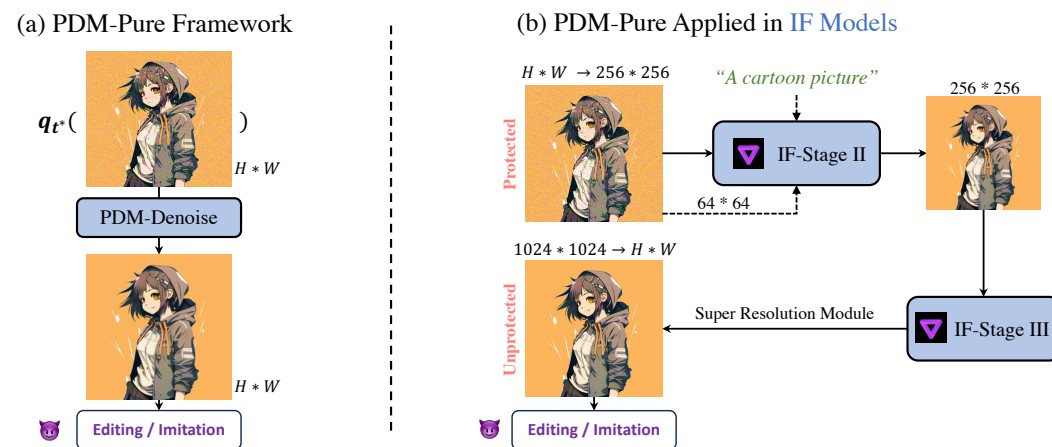

Figure 5: **PDM-Pure is Easy to Design:** (a) PDM-Pure applies SDEdit Meng et al. (2021) in the pixel space: it first runs forward diffusion with a small step $t^*$ and then runs the denoising process. (b) We adapt the framework to DeepFloyd-IF Shonenkov et al., one of the strongest PDMs. PDM-Pure can effectively remove strong protective perturbations (e.g. $\delta = 16/255$). The images we tested are sized $512 \times 512$.

patterns generated by existing methods in Section 5. This new landscape poses new challenges to ensure the security and robustness of diffusion-based copyright protection techniques.

## 5 PDM-PURE: PDM AS A STRONG UNIVERSAL PURIFIER

Since PDMs are robust to adversarial perturbations, a natural idea emerges: we can utilize PDMs as a universal purification network. This approach could potentially eliminate any adversarial patterns without knowing the nature of the attacks. We term this framework **PDM-Pure**, which is a general framework to deal with all the perturbations utilized nowadays. To fully harness the capabilities of PDM-Pure, we need to fulfill two basic requirements: (1) The perturbation adds an out-of-distribution pattern as reflected in existing works on adversarial purification/attacks using diffusion models (Nie et al., 2022; Xue et al., 2024) (2) The PDM being used is strong enough to represent $p(x_0)$, which can be largely determined by the dataset they are trained on.

It is **effortless** to design a PDM-Pure. The key idea behind this method is to run SDEdit in the pixel space. Given any strong pixel-space diffusion model, we add a small noise to the protected images and run the denoising process (Figure 5), and then the adversarial pattern should be removed. The key idea of PDM-Pure is simple. In practice, we need to adjust the pipeline to fit the resolution of the PDMs being used.

In the main paper, we adopt DeepFloyd-IF (Shonenkov et al.), the strongest pixel-space diffusion models nowadays as the purifier. We conduct experiments on purifying protected images sized $512 \times 512$. For images with a larger resolution, purifying in the resolution of $256 \times 256$ may lose information. In Appendix I we show that PDM-Pure can also be applied to purify patches of high-resolution inputs, removing widely used protections like Glaze on artworks. More details about the how we run DeepFloyd-IF as the purification pipeline are in the Appendix G.

## 6 EXPERIMENTS

In this section, we conduct experiments with various attacking methods and models to support the following two conclusions:

- **(C1)**: PDMs are much more adversarially robust than LDMs, and PDMs can not be effectively attacked using all the existing attacks for LDMs.

| Methods | AdvDM | AdvDM(-) | SDS(-) | SDS(+) | SDST | Photoguard | Mist | Mist-v2 |
|---|---|---|---|---|---|---|---|---|
| Before Protection | 166 | 166 | 166 | 166 | 166 | 166 | 166 | 166 |
| After Protection | 297 | 221 | 231 | 299 | 322 | 375 | 372 | 370 |
| Crop-Resize | 210 | 271 | 228 | 217 | 280 | 295 | 289 | 288 |
| JPEG | 296 | 222 | 229 | 297 | 320 | 359 | 351 | 348 |
| Adv-Clean | 243 | 201 | 204 | 244 | 243 | 266 | 282 | 270 |
| LDM-Pure | 300 | 251 | 235 | 300 | 350 | 385 | 380 | 375 |
| GrIDPure | 200 | 182 | 195 | 200 | 210 | 220 | 230 | 210 |
| PDM-Pure (ours) | **161** | **170** | **165** | **159** | **179** | **175** | **178** | **170** |

Table 2: **Quantiative Measurement of Different Purification Methods in Different Scale (FID-score)**: We compute the FID-score of edited purified images over the clean dataset. PDM-Pure achieves the best results on all protection methods, under strong protection with $\delta = 16$. GrID-Pure Zhao et al. (2023) can also perform reasonably, but the performance is limited because the PDM they used is not strong enough.

- **(C2)**: PDMs can be applied to effectively purify all of the existing protective perturbations. Our PDM-Pure based on DeepFloyd-IF shows state-of-the-art purification power.

## 6.1 MODELS, DATASETS, AND METRICS

The models we used can be categorized into LDMs and PDMs. For LDMs, we use Stable Diffusion V-1.4, V-1.5 (SD-V-1.4, SD-V-1.5) (Rombach et al., 2022), and Diffusion Transformer (DiT-XL/2) (Peebles and Xie, 2023), and for PDMs we use Guided Diffusion (GD) (Dhariwal and Nichol, 2021) trained on ImageNet (Deng et al., 2009), and DeepFloyd Stage I and Stage II (Shonenkov et al.).

For models trained on the ImageNet (DiT, GD), we run adversarial attacks and purification on a 1k subset of the ImageNet validation dataset. For models trained on LAION, we run tests on the dataset proposed in (Xue et al., 2023), which includes 400 cartoon, artwork, landscape, and portrait images.

For protection methods, we consider almost all the representative approaches, including AdvDM (Liang et al., 2023), SDS (Xue et al., 2023), Mist (Liang and Wu, 2023), Mist-v2 (Zheng et al., 2023), Photoguard (Salman et al., 2023) and Glaze (Shan et al., 2023). We also test the methods in the design space proposed in (Xue et al., 2023), including SDS(-), AdvDM(-), and SDST. In contrast to other existing methods, they are based on gradient descent and have shown great performance in deceiving LDMs.

We measure the SDEdit results after the adversarial attacks using Fréchet Inception Distance (FID) (Heusel et al., 2017) over the relevant datasets (for models trained on ImageNet such as GD (Dhariwal and Nichol, 2021) and DiT (Peebles and Xie, 2023) we use a sub-dataset of ImageNet as the relevant dataset, for those trained on LAION, we use the collected dataset in (Xue et al., 2023) to calculate the FID). We also use Image-Alignment Score (IA-score) (Kumari et al., 2023), which can be used to calculate the cosine-similarity between the CLIP embedding of the edited image and the original image. Also, we use some basic evaluations, where we calculate the Structural Similarity (SSIM) (Wang et al., 2004) and Perceptual Similarity (LPIPS) (Zhang et al., 2018) compared with the original images.

All the experiments are written using PyTorch and run in the Linux system, and all of them can be conducted on four A6000 GPUs.

## 6.2 (C1) DIFFUSION DENOISING PROCESS IS MORE ROBUST THAN WE THINK

In Table 1, we attack different LDMs and PDMs with one of the most popular adversarial losses (Zheng et al., 2023) in Equation 1, which can be interpreted as fooling the denoiser using a Monte-Carlo-based loss. Given the attacked samples, we test the SDEdit results on the attacked samples, which can be generally used to test whether the samples are adversarial for the diffusion model or not. We use FID-score (Heusel et al., 2017), SSIM (Wang et al., 2004), LPIPS (Zhang et al.,

2018), and IA-Score (Kumari et al., 2023) to measure the quality of the attack. If the quality of the generated images decreases a lot compared to edited clean images, then the attack is successful. We found that for all LDMs, attacks using the adversarial loss successfully provide protection. However, for all PDMs, the adversarial attacks do not work. This phenomenon occurs across all scales of perturbation. For example, when the FID of LDMs increased by over 100, the FID of PDMs remained nearly unchanged. We also show some visualizations in Figure 2, which illustrate that the perturbation will affect the LDMs but not the PDMs.

To further investigate how robust the PDM is, we test other advanced attacking methods, including the End-to-End Diffusion Attacks (E2E-Photoguard) proposed in (Salman et al., 2023) and the Improved Targeted Attack (ITA) proposed in (Zheng et al., 2023). Though the End-to-End attack is usually impractical to run, it shows the strongest performance when attacking LDMs. We find that both attacks are not successful in PDM settings. We show attacked samples and edited samples in Figure 2, 3, 4 as well as the Appendix H. In conclusion, existing adversarial attack methods for diffusion models can only work for LDMs, and PDMs are more robust than we think.

### 6.3 (C2) PDM-PURE: A UNIVERSAL PURIFIER THAT IS SIMPLE YET EFFECTIVE

PDM-Pure is simple: we just run SDEdit to purify the protected image in the pixel space. Given our assumption that PDMs are quite robust, we can use PDMs trained on large-scale datasets as a universal black-box purifier. We follow the model pipeline introduced in Section 5 and purify images protected by various methods as shown in Table 2.

PDM-Pure is effective: from Table 2 we can see that the purification will remove adversarial patterns for all the protection methods we tested, largely decreasing the FID score for the SDEdit task. Also, we test the protected images and purified images in more tasks including Image Inpainting (Song et al., 2020), Textual-Inversion (Gal et al., 2022), and LoRA customization (Hu et al., 2021). We show purification results for inpainting in Figure 12, and purification results for LoRA in Figure 7. We show more results in Figure 16 in the appendix.

Both qualitative and quantitative results show that the purified images are no longer adversarial and can be effectively edited or imitated in different tasks without any obstruction.

Also, PDM-Pure shows SOTA results compared with previous purification methods, including some simple purifiers based on compression and filtering like Adv-Clean, crop-and-resize, JPEG Compression, and SDEdit-based methods like GrIDPure (Zhao et al., 2023), which uses patchified SDEdit with a GD (Dhariwal and Nichol, 2021). We also add LDM-Pure as a baseline to show that LDMs can not be used to purify the protected images. For GrIDPure, we use Guided-Diffusion trained on ImageNet to run patchified purification. All the experiments are conducted on the datasets collected in (Xue et al., 2023) under the resolution of $512 \times 512$. Results for higher resolutions are presented in Appendix I. We also test the ablation of timesteps used for PDM-Pure in Appendix Appendix J, from which we can see the sweet point of timesteps: $t^*$ around $0.15$ works well. We also find that PDM-Pure works better for cartoon pictures with larger plain color patches. For pictures with many details like oil paintings, it will lose some detail; however, generally the art style can still be learned well by LoRA from the attacker's perspective (e.g. Claude Monet-style in Appendix Figure 13 ).

## 7 CONCLUSIONS AND FUTURE DIRECTIONS

In this paper, we present novel insights that while many studies demonstrate the ease of finding adversarial samples for Latent Diffusion Models (LDMs), Pixel Diffusion Models (PDMs) exhibit far greater adversarial robustness than previously assumed. We are the first to investigate the adversarial samples for PDMs, revealing a surprising discovery that existing attacks fail to fool PDMs. Leveraging this insight, we propose utilizing strong PDMs as universal purifiers, resulting in PDM-Pure, a simple yet effective framework that can purify protective perturbations in a black-box manner.

Pixel is a barrier for real protection against adversarial attacks. Since PDMs are quite robust, they cannot be easily attacked. PDMs can even be used to purify the protective perturbations, challenging the current assumption for the safe protection of generative diffusion models. We advocate rethinking the problem of adversarial samples for generative diffusion models and unauthorized image protection

based on it. More rigorous studies need to be conducted to better understand the mechanism behind the robustness of PDMs. Furthermore, we can utilize it as a new structure for many other tasks.

## 8 LIMITATIONS

In this paper, we present empirical insights demonstrating the robustness of PDMs by attacking various PDMs using different methods. We do not provide a theoretical analysis of the underlying mechanisms. For PDM-Pure, though the purification is stronger than previous methods, there is still a trade-off between purifying power and the preservation of image details, particularly in images with intricate details. Additionally, we rely on patched purification for larger images, which may result in subtle edge shadows between patches.

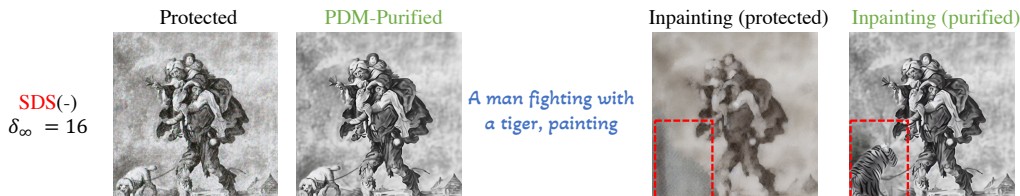

Figure 6: **PDM-Pure makes the Protected Images no longer Protected**: PDMs can help effectively remove adversarial patterns to bypass the protection for LDMs, here we show an example on inpainting with SDS protection proposed in (Xue et al., 2023). We put more results on more attacks and more examples in the Appendix Figure 16.

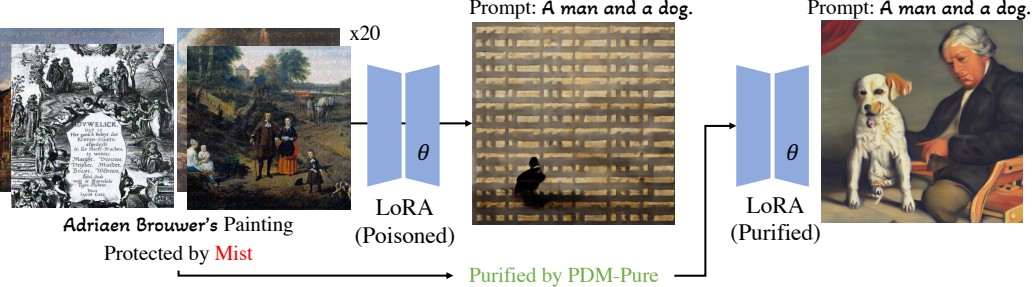

Figure 7: **PDM-Pure makes the Protected Images no longer LoRA-proof**: PDMs can also help effectively remove adversarial patterns to bypass the protection for LDMs under LoRA settings. Here we use Mist (Liang and Wu, 2023) to perturb the images. We put more results on more attacks and more examples in the Appendix Figure 16.

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

# Appendix

## A  BROADER IMPACT

We present significant insights in two crucial areas: adversarial machine learning research on generative diffusion models, and the protection of copyright against the malicious use of diffusion models. While existing works have revealed the vulnerability of latent diffusion models, we show that the general diffusion model in the pixel space is quite robust. PDMs reveal two new threats to the safety application of diffusion models: (1) since PDMs are robust and no existing perturbation can effectively attack them, it means that copyright protection against PDMs cannot be easily achieved with existing protective perturbations (2) PDMs can be used to purify the protective noise used to protect the LDMs, meaning that the current protection for LDMs can be bypassed. We still have a long way to go to achieve good protection against diffusion models, and more efforts should be dedicated to enhancing copyright protection for PDMs and making current protective measures more robust and reliable.

## B  DETAILS ABOUT DIFFERENT DIFFUSION MODELS IN THIS PAPER

Here we introduce the diffusion models used in this work, which cover different types of diffusion (LDM, PDM), different training datasets, different resolutions, and different model structures (U-Net, Transformer):

**Guided Diffusion (PDM)**  We use the implementation and checkpoint from `https://github.com/openai/guided-diffusion`, the Guided Diffusion models we used are trained on ImageNet (Deng et al., 2009) in resolution $256 \times 256$, the editing results are tested on sub-dataset of ImageNet validation set sized 500.

**IF-Stage I (PDM)**  This is the first stage of the cascaded DeepFloyd IF model (Shonenkov et al.) from `https://github.com/deep-floyd/IF`. It is trained on LAION 1.2B with text annotation. It has a resolution of $64 \times 64$. the editing results are tested on the image dataset introduced in (Xue et al., 2023), including 400 anime, portrait, landscape, and artwork images.

**IF-Stage II (PDM)**  This is the second stage of the cascaded DeepFloyd IF model (Shonenkov et al.) from `https://github.com/deep-floyd/IF`. It is a conditional diffusion model in the pixel space with $256 \times 256$, which is conditioned on $64 \times 64$ low-resolution images. During the attack, we freeze the image condition and only attack the target image to be edited.

**Stable Diffusion V-1.4 (LDM)**  It is one of the most popular LDMs from `https://huggingface.co/CompVis/stable-diffusion-v1-4`, also trained on text-image pairs, which has been widely studied in this field. It supports resolutions of $256 \times 256$ and $512 \times 512$, both can be easily attacked. The encoder first encodes the image sized $H \times W$ into the latent space sized $4 \times H/4 \times W/4$, and then uses U-Net combined with cross-attention to run the denoising process.

**Stable Diffusion V-1.5 (LDM)**  It has the same structure as Stable Diffusion V-1.4, which is also stronger since it is trained with more steps, from `https://huggingface.co/runwayml/stable-diffusion-v1-5`.

**DiT-XL (LDM)**  It is another popular latent diffusion model, that uses the backbone of the Transformer instead of the U-Net. We use the implementation from the original repository `https://github.com/facebookresearch/DiT/`.

# C DETAILS ABOUT DIFFERENT PROTECTION METHODS IN THIS PAPER

We introduce different protection methods tested in this paper, of which all the original versions are designed for LDMs. All the adversarial attacks work under white box settings of PGD-attack, varying from each other with different adversarial losses:

**AdvDM** AdvDM is one of the first adversarial attacks proposed in (Liang et al., 2023), it used a Monte-Carlo-based adversarial loss which can effectively attack latent diffusion models, we also call this loss semantic loss:

$$\mathcal{L}_S(x) = \mathbb{E}_{t,\epsilon}\mathbb{E}_{z_t \sim q_t(\mathcal{E}_\phi(x))}\|\epsilon_\theta(z_t, t) - \epsilon\|_2^2 \tag{3}$$

**PhotoGuard** PhotoGuard is proposed in (Salman et al., 2023), it takes the encoder, making the encoded image close to a target image $y$, we also call it textural loss:

$$\mathcal{L}_T(x) = -\|\mathcal{E}_\phi(x) - \mathcal{E}_\phi(y)\|_2^2 \tag{4}$$

**Mist** Mist (Liang and Wu, 2023) finds that $L_T(x)$ can better enhance the attacks if the target image $y$ is chosen to be periodical patterns, the final loss combined $L_T(x)$ and $L_S(x)$:

$$\mathcal{L} = \lambda L_T(x) + L_S(x) \tag{5}$$

**SDS(+)** Proposed in (Xue et al., 2023), it is proven to be a more effective attack compared to the original AdvDM, where the gradient $\nabla_x \mathcal{L}(x)$ is expensive to compute. By using the score distillation-based loss, it shows good performance and remains effective at the same time:

$$\nabla_x \mathcal{L}_{SDS}(x) = \mathbb{E}_{t,\epsilon}\mathbb{E}_{z_t}\left[\lambda(t)(\epsilon_\theta(z_t, t) - \epsilon)\frac{\partial z_t}{\partial x_t}\right] \tag{6}$$

**SDS(-)** Similar to SDS(+), it swaps gradient ascent in the original PGD with gradient descent, which turns out to be even more effective.

$$\nabla_x \mathcal{L}_{SDS(-)}(x) = -\mathbb{E}_{t,\epsilon}\mathbb{E}_{z_t}\left[\lambda(t)(\epsilon_\theta(z_t, t) - \epsilon)\frac{\partial z_t}{\partial x_t}\right] \tag{7}$$

**Mist-v2** It was proposed in (Zheng et al., 2023) using the Improved Targeted Attack (ITA), which turns out to be very effective, especially when the budget is small. It is also more effective to attack LoRA:

$$\mathcal{L}_S(x) = \mathbb{E}_{t,\epsilon}\mathbb{E}_{z_t \sim q_t(\mathcal{E}_\phi(x))}\|\epsilon_\theta(z_t, t) - z_0\|_2^2 \tag{8}$$

where $z_0 = \mathcal{E}(y)$ is the latent of the target image, which is the same as the typical image used in Mist.

**Glaze** It is the most popular protection claimed to safeguard artists from unauthorized imitation (Shan et al., 2023) and is widely used by the community. while it is not open-sourced, it also attacks the encoder like the Photoguard. Here we only test it in the purification stage, where we show that the protection can also be bypassed.

**End-to-End Attack** It is also first proposed in (Salman et al., 2023), which attacks the editing pipeline in a end-to-end manner. Although it is strong, it is not practical to use and does not show dominant privilege compared with other protection methods.

# D    DETAILS ABOUT THE LATENT ATTACKS FOR PDMS

In an attempt to extend the latent-space attacks onto PDMs, (Shih et al., 2024) introduces atkPDM+. This method uses a pre-trained VAE to attack the PDM by extracting feature vectors from the encoder network. The attack optimizes the latent vector with a Wasserstein distance objective calculated at the VAE middle layer activations:

$$\mathcal{L}_{attack}(x_t, x_t^{adv}) = -\mathcal{W}_2(\mathcal{U}_\theta^{(mid)}(x_t), \mathcal{U}_\theta^{(mid)}(x_t^{adv}))$$

A second optimization cycle is then run to limit the change in pixel-space by optimizing the distance between the feature vector generated by a pre-trained image classifier taken from the original image and the decoded attacked latent.

We observe, however, that in this attack the perturbation is clearly visible, and the pixel-wise distance is large: $\|x - x_{adv}\| \geq 150$.

# E    DETAILS ABOUT THE EVALUATION METRICS

Here we introduce the quantitative measurement we used in our experiments:

- We measure the SDEdit results after the adversarial attacks using Fréchet Inception Distance (FID) (Heusel et al., 2017) over the relevant datasets (for models trained on ImageNet such as GD (Dhariwal and Nichol, 2021) and DiT (Peebles and Xie, 2023) we use a sub-dataset of ImageNet as the relevant dataset, for those trained on LAION, we use the collected dataset to calculate the FID). We also use Image-Alignment Score (IA-score) (Kumari et al., 2023), which can be used to calculate the cosine-similarity between the CLIP embedding of the edited image and the original image. Also, we use some basic evaluations, where we calculate the Structural Similarity (SSIM) (Wang et al., 2004) and Perceptual Similarity (LPIPS) (Zhang et al., 2018) compared with the original images.

- To measure the purification results, we test the Fréchet Inception Distance (FID) (Heusel et al., 2017) over the collected dataset compared with the dataset generated by running SDEdit over the purified images in the strength of $0.3$.

# F    DETAILS ABOUT DIFFERENT PURIFICATION METHODS

**Adv-Clean:**  https://github.com/lllyasviel/AdverseCleaner, a training-free filter-based method that can remove adversarial noise for a diffusion model, it works well to remove high-frequency noise.

**Crop & Resize:**  first crops the image by $20\%$ and then resizes the image to the original size, it turns out to be one of the most effective defense methods (Liang and Wu, 2023).

**JPEG compression:**  (Sandoval-Segura et al., 2023) reveals that JPEG compression can be a good purification method, and we adopt the $65\%$ as the quality of compression in (Sandoval-Segura et al., 2023).

**LDM-Pure:**  We also try to use LDMs to run SDEdit as a naive purifier, sadly it does not work, because the adversarial protection transfers well between different LDMs.

**GrIDPure:**  It is proposed in (Zhao et al., 2023) as a purifier, GrIDPure first divides an image into patches sized $128 \times 128$, and then purifies the 9 patches sized $256 \times 256$. Also, it combined the four corners sized $128 \times 128$ to purify it so we have 10 patches to purify in total. After running SDEdit with a small noise (set to $0.1T$), we reassemble the patches into the original size, pixel values are assigned using the average values of the patches they belong to. More details can be seen in (Zhao et al., 2023).

# G  DETAILS ABOUT PDM-PURE

Here, we explain in detail how to adapt DeepFloyd-IF (Shonenkov et al.), the strongest open-source PDM as far as we know, for PDM-Pure. DeepFloyd-IF is a cascaded text-to-image diffusion model trained on 1.2B text-image pairs from LAION dataset (Schuhmann et al., 2022). It contains three stages named IF-Stage I, II, and III. Here we only use Stage II and III since Stage I works in a resolution of $64$ which is too low. Given a perturbed image $x_{W \times H}$ sized $W \times H$, we first resize it into $x_{64 \times 64}$ and $x_{256 \times 256}$. Then we use a general prompt $\mathcal{P}$ to do SDEdit (Meng et al., 2021) using the Stage II model:

$$x_t = \textbf{IF-II}(x_{t+1}, x_{64 \times 64}, \mathcal{P}) \tag{9}$$

where $t = T_{\text{edit}} - 1, ..., 1, 0$, $x_{T_{\text{edit}}} = x_{256 \times 256}$. A larger $T_{\text{edit}}$ may be used for larger noise. $x_0$ is the purified image we get in the $256 \times 256$ resolution space, where the adversarial patterns should be already purified. We can then use IF Stage III to further up-sample it into $1024 \times 1024$ with $x_{1024 \times 1024} = \textbf{IF-III}(x_0, p)$. Finally, we can sample into $H \times W$ as we want through downsampling. This whole process is demonstrated in Figure 5. After purification, the image is no longer adversarial to the targeted diffusion models and can be effectively used in downstream tasks.

# H  MORE EXPERIMENTAL RESULTS

In this section, we present more experimental results.

## H.1  MORE VISUALIZATIONS OF ATTACKING PDMS

We show more results of attacking LDMs and PDMs in Figure 8, where we attack them with a different budget $\delta = 4, 8, 16$. We can see that all the LDMs can be easily attacked, while the PDMs cannot be attacked, even the largest perturbations will not fool the editing process. In fact, the editing process is trying to purify the strange perturbations.

## H.2  MORE VISUALIZATIONS OF PDM-PURE AND BASELINE METHODS

We show more qualitative results of the proposed PDM-Pure based on IF. First, we show purified samples of PDM-Pure in Figure. 10, from which we can see that PDM-Pure can remove large protective perturbations and largely preserve details.

Compared with GrIDPure (Zhao et al., 2023), we find that PDM-Pure shows better results when the noise is large and colorful, as is illustrated in Figure 11. Also, though GrIDPure merges patches, it still shows boundary lines between patches.

Compared with other baseline purification methods such as Adv-Clean, Crop-and-Resize, and JPEG compression, PDM-Pure shows much better results (Figure 9) for different kinds of protective noise, showing that it is capable to serve as a universal purifier. We choose AdvDM, Mist, and SDS as the representative of three kinds of protection.

## H.3  MORE VISUALIZATIONS OF PDM-PURE FOR DOWNSTREAM TASKS

After applying PDM-Pure to the protected images, they are no longer adversarial to LDMs and can be easily edited or imitated. Here we will demonstrate more results on editing the purified images on downstream tasks.

In Figure 12, we show more results to prove that the purified images can be edited easily, and the quality of the editing results is high. It means that PDM-Pure can bypass the protection very well for inpainting tasks.

In Figure 13 we show more results on purifying Mist (Liang and Wu, 2023) and Glaze (Shan et al., 2023) perturbations, and then running LoRA customized generation. From the figure, we can see that PDM-Pure can make the protected images easy to imitate again.

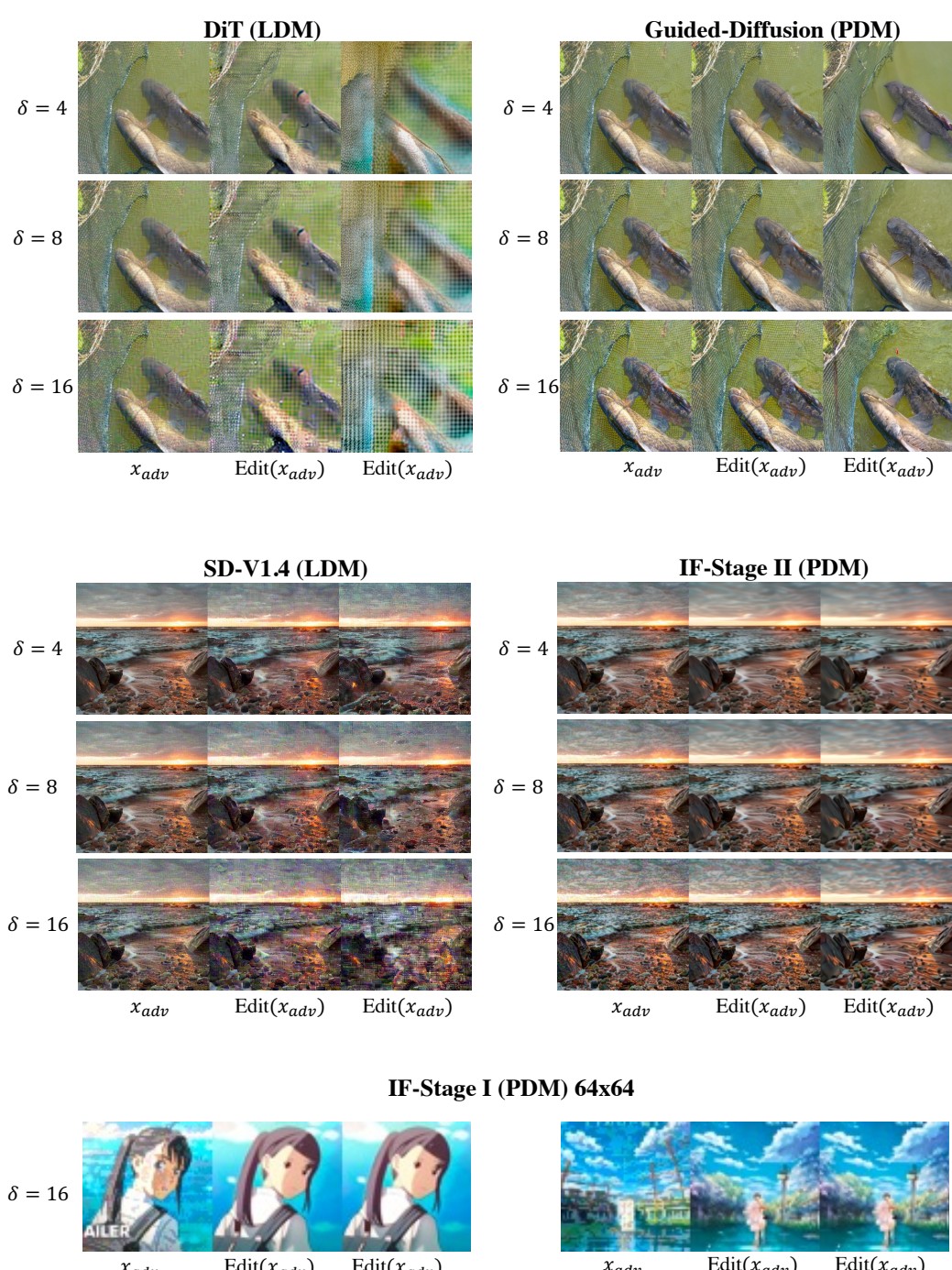

Figure 8: **PDMs cannot be Attacked as LDMs**: we conduct experiments on various models with various budgets, even the largest budget will not affect the PDMs, showing that PDMs are adversarially robust. For each block, the first column is the attacked image, and the second and third columns are edited images, where the third column adopts larger editing strength.

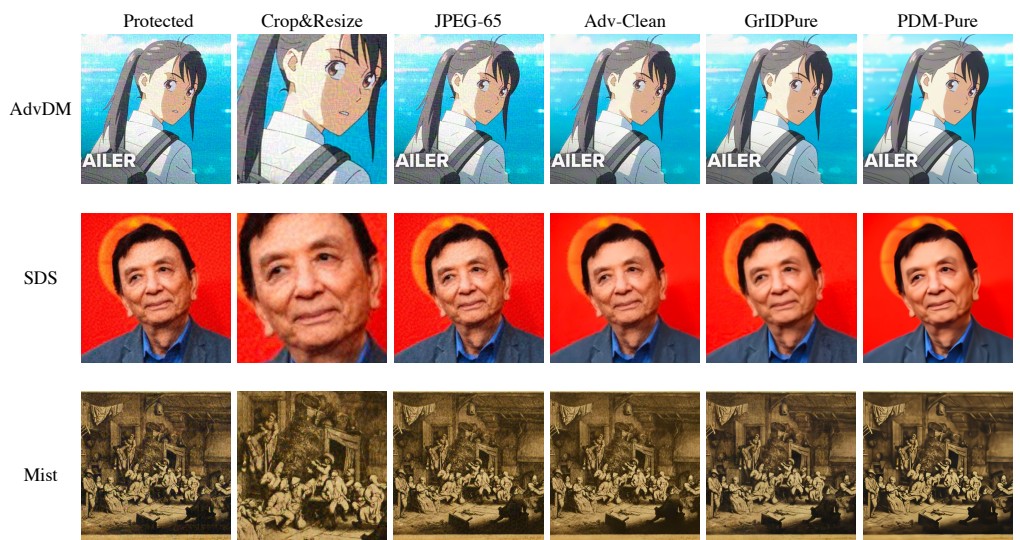

Figure 9: **PDM-Pure Compared With Other Baseline Methods**: we test all the baselines on three typical kinds of protection methods, with $\delta = 16/255$. PDM-Pure shows strong performance.

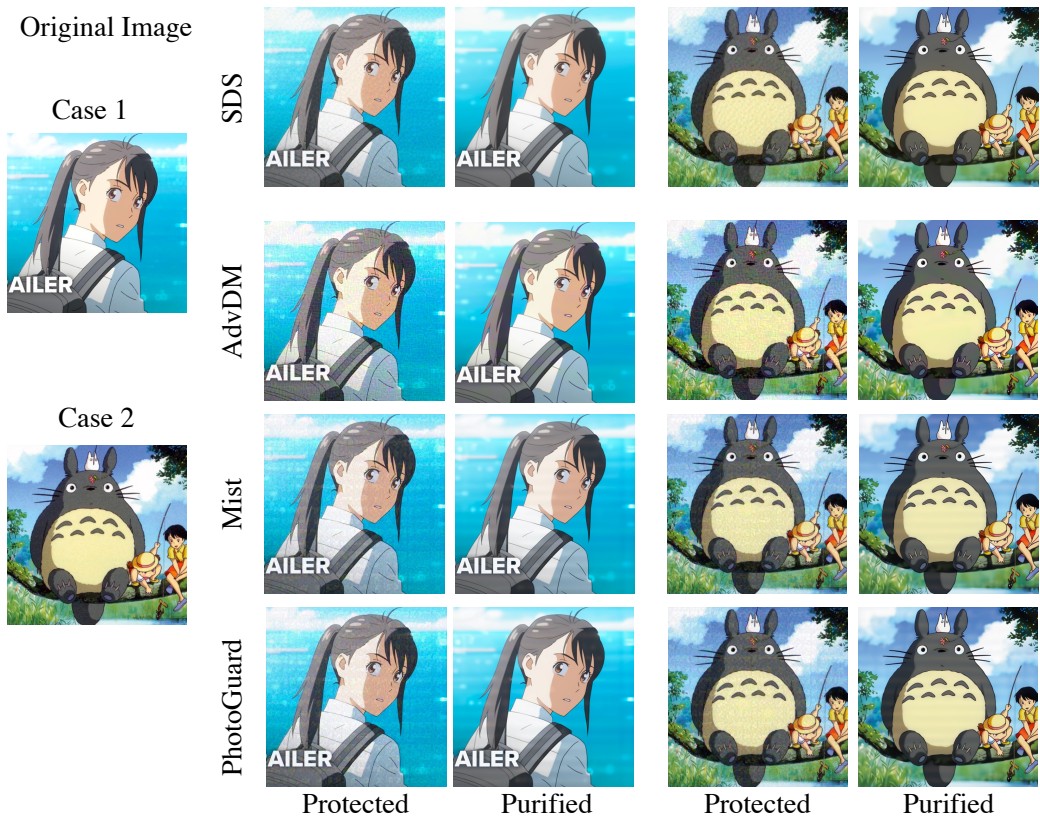

Figure 10: **More Purification Results of PDM-Pure**: we show purification results compared with the clean image, working on SDS, AdvDM, Mist, and PhotoGuard.

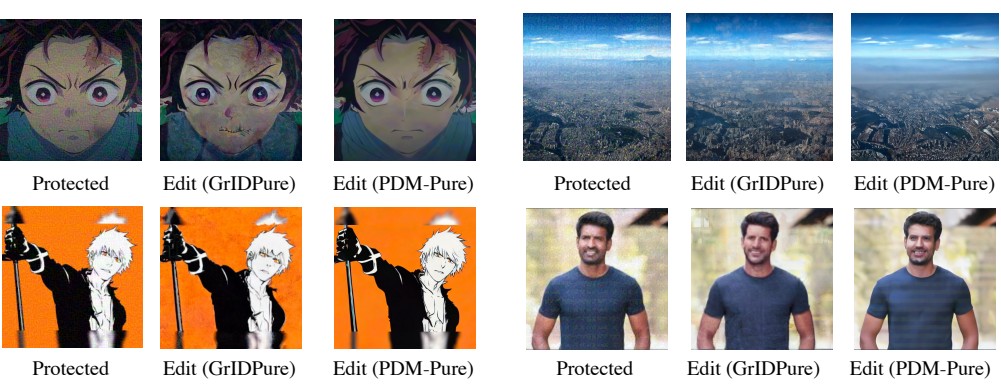

Figure 11: **PDM-Pure vs GrIDPure**: PDM-Pure is better than GrIDPure, especially when the adversarial pattern is strong such as AdvDM. The bottom half of this figure shows the editing results of purified images, we can see that the editing results of GrIDPure still have some artifacts.

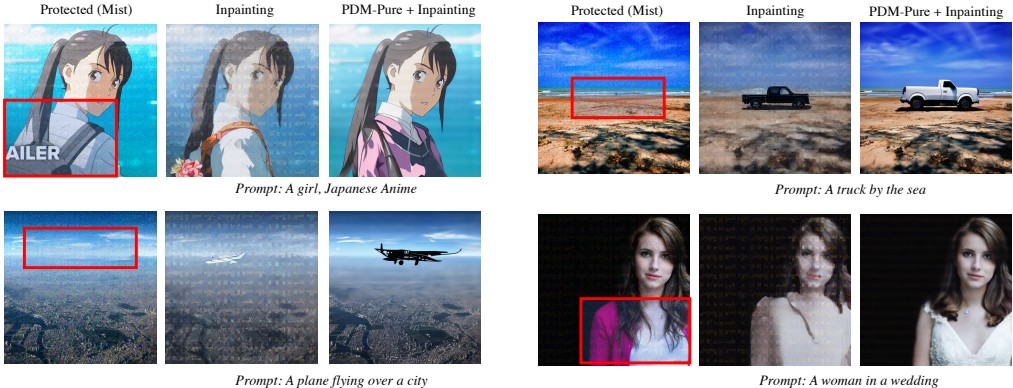

Figure 12: **More Results of PDM-Pure Bypassing Protection for Inpainting**: after purification, the protected images can be easily inpainted with high quality. The protective perturbations are generated using Mist with $\delta = 16/255$, which is a strong perturbation.

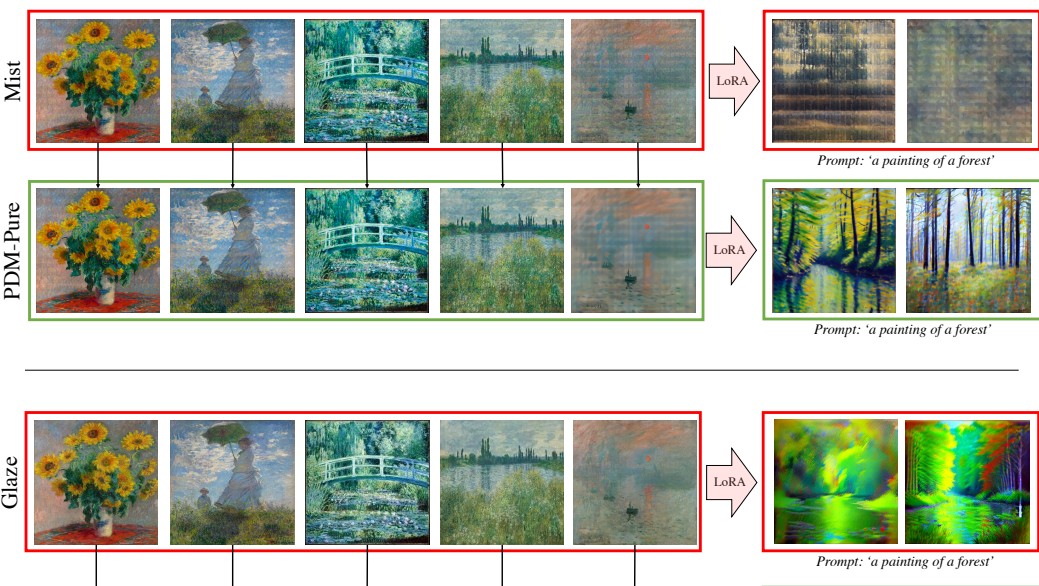

Figure 13: **More Results of PDM-Pure Bypassing Protection for LoRA**: after purification, the protected images can be imitated again. Here we show examples using 5 paintings of Claude Monet.

# I  PDM-PURE FOR HIGHER RESOLUTION

In this paper, we mainly apply PDM-Pure for images sized $512 \times 512$, which is also the most widely used resolution for latent diffusion models. When the resolution is $512 \times 512$, running SDEdit using Stage II of DeepFloyd makes sense, while if the image size becomes larger, details may be lost because of the downsampling. Hopefully, we can still do purification patch-by-patch with PDM-Pure, in Figure 14 we show purification results on images with different resolutions protected by Glaze (Shan et al., 2023).

# J  ABLATIONS OF $t^*$ IN PDM-PURE

The PDM-Pure on DeepFloyd-IF we used in this paper uses the default settings of SDEdit with $t^* = 0.1T$. And we respace the diffusion model into 100 steps, so we only need to run 10 denoising steps. It can be run on one A6000 GPU, occupying $22G$ VRAM in 30 seconds.

Here we show some ablation about the choice of $t^*$. In fact, in many SDEdit papers, $t^*$ can be roughly defined by trying different $t^*$ that can be used to purify different levels of noise. We try $t^* = 0.01, 0.1, 0.2$, in Figure 15 we can see that when $t^* = 0.01$ the noise is not fully purified, and when $t^* = 0.2$, the details in the painting are blurred. It should be noted that the sweet spot for different images and different noises can be slightly different, so one is advised to do some trials before purification.

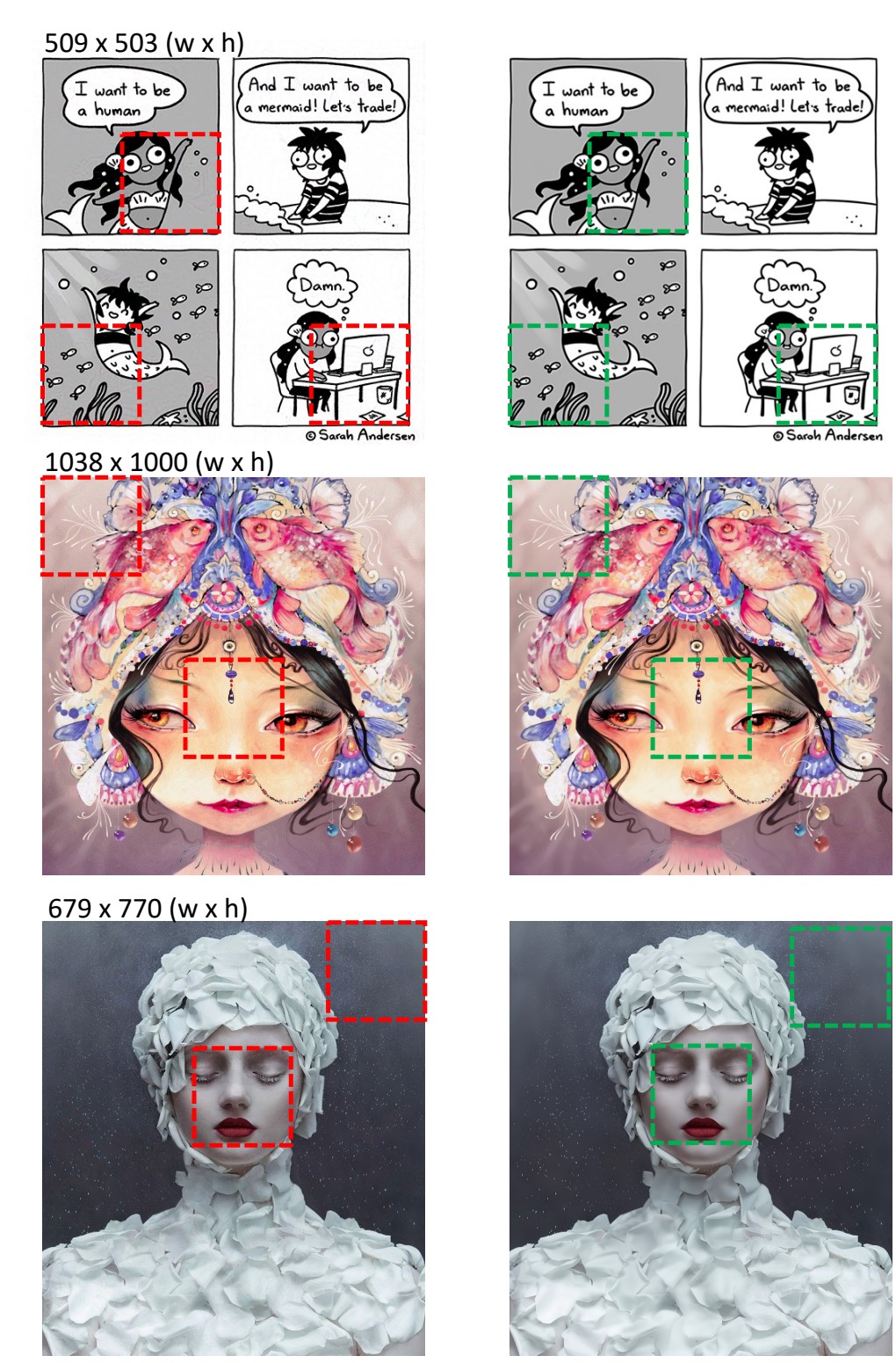

Figure 14: **PDM-Pure Working On Images with Higher Resolution**: we show the results of applying PDM-Pure for images with higher resolutions, the images are protected using Glaze (Shan et al., 2023). We can see from the figure that the adversarial patterns (in the red box) can be effectively purified (in the green box). Zoom in on the computer for a better view.

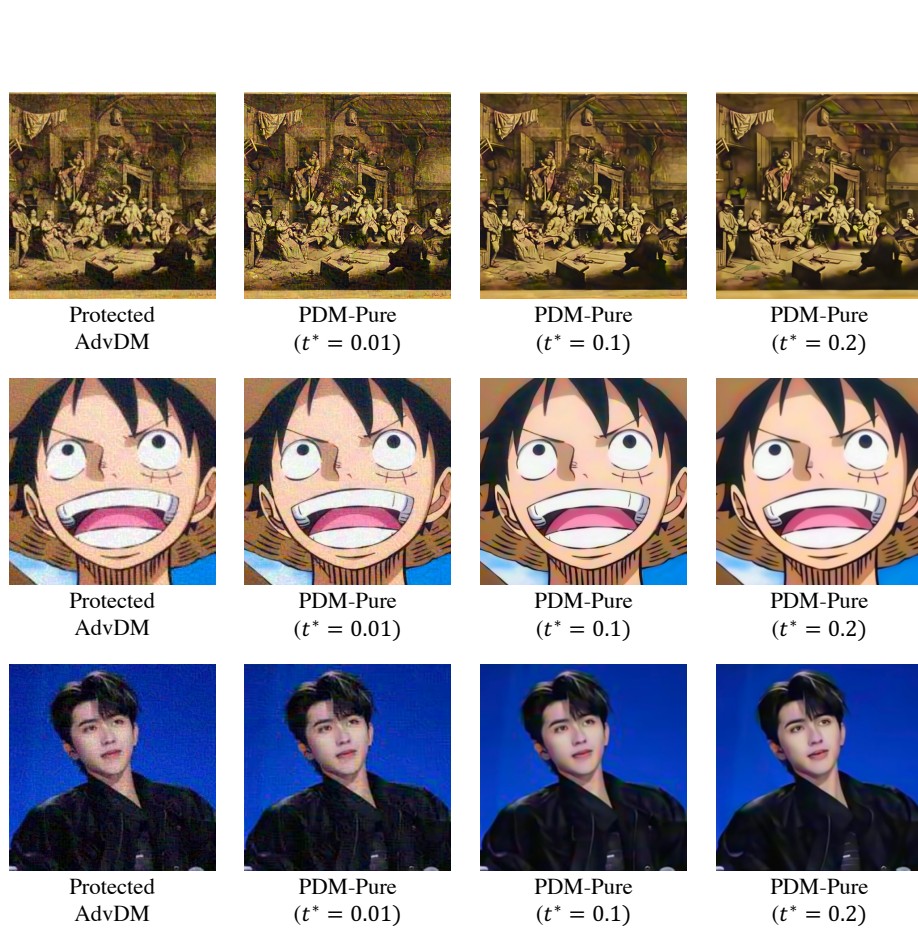

Figure 15: **PDM-Pure with Different** $t^*$

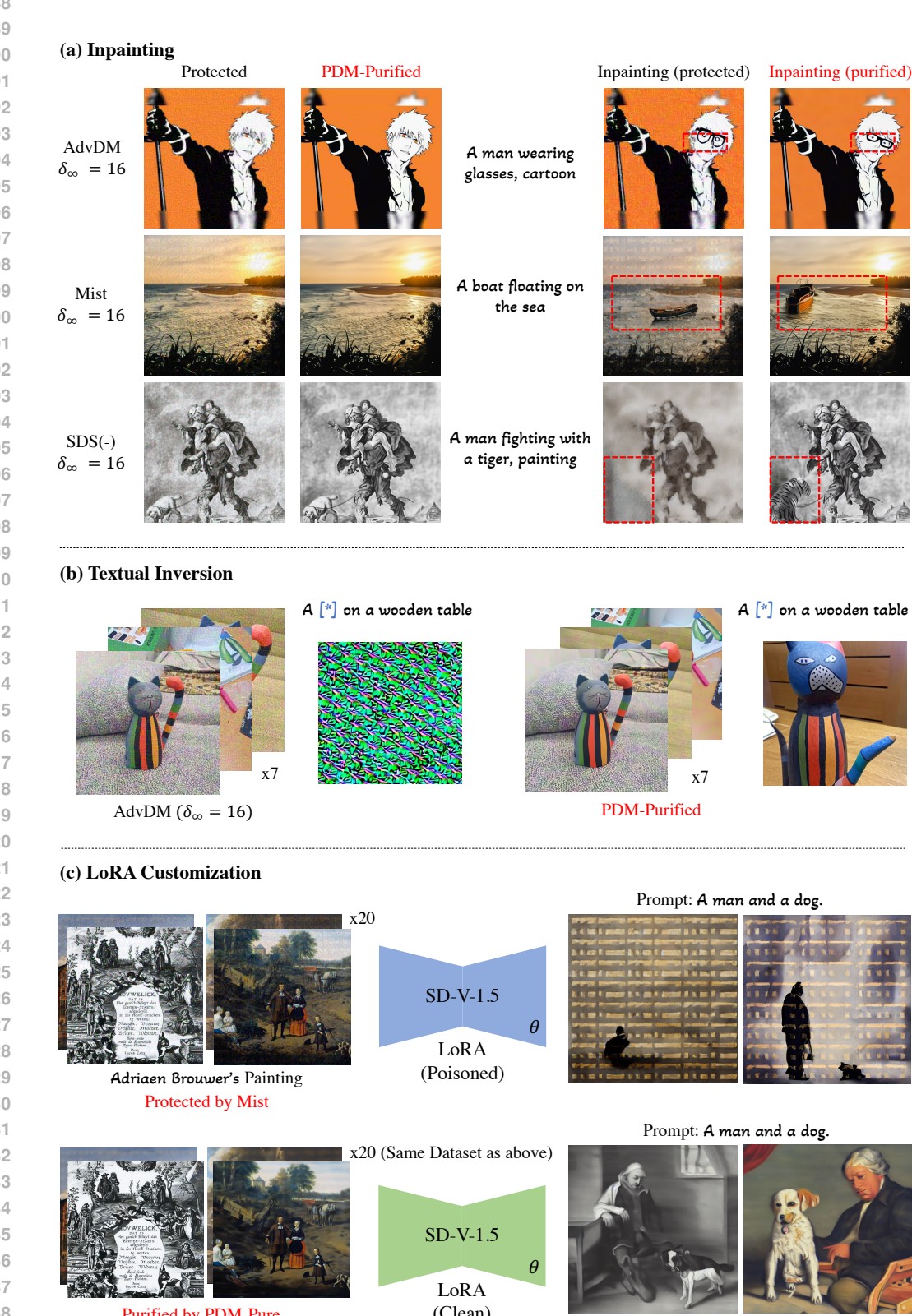

Figure 16: **PDM-Pure for inpainting, textual inversion and LoRA**

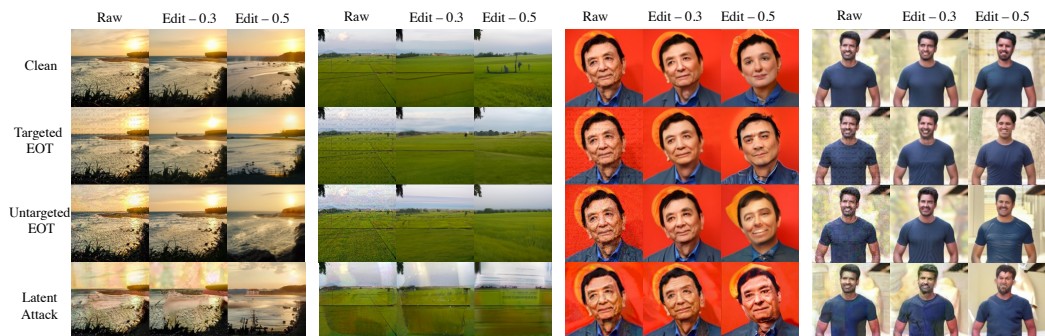

Figure 17: **More results for adaptive attacks for PDM**: here we show attacking results for one PDM (Guided-Diffusion (Dhariwal and Nichol, 2021)), we conduct SDEdit with two different strengths 0.3 and 0.5 to test the attacking performance. We show results for targeted/untargeted attack with gradient aggregation (Targeted/Untargeted EOT), we also show results for latent attacks following the settings in (Shih et al., 2024). We can see all the attacks is not that successful for the pixel-space diffusion model.

| Methods | AdvDM | AdvDM(-) | SDS(-) | SDS(+) | SDST | Photoguard | Mist | Mist-v2 |
|---------|-------|----------|--------|--------|------|------------|------|---------|
| Clean | 0.95 | 0.95 | 0.95 | 0.95 | 0.95 | 0.95 | 0.95 | 0.95 |
| Attacked | 0.73 | 0.70 | 0.68 | 0.76 | 0.61 | 0.61 | 0.62 | 0.63 |
| PDM-Pure | 0.94 | 0.93 | 0.92 | 0.93 | 0.93 | 0.94 | 0.93 | 0.93 |

Table 3: **IA Score of SDEdit results After Purification**

Protected Image    DiffPure + SDXL    SD-X4-Upscaler

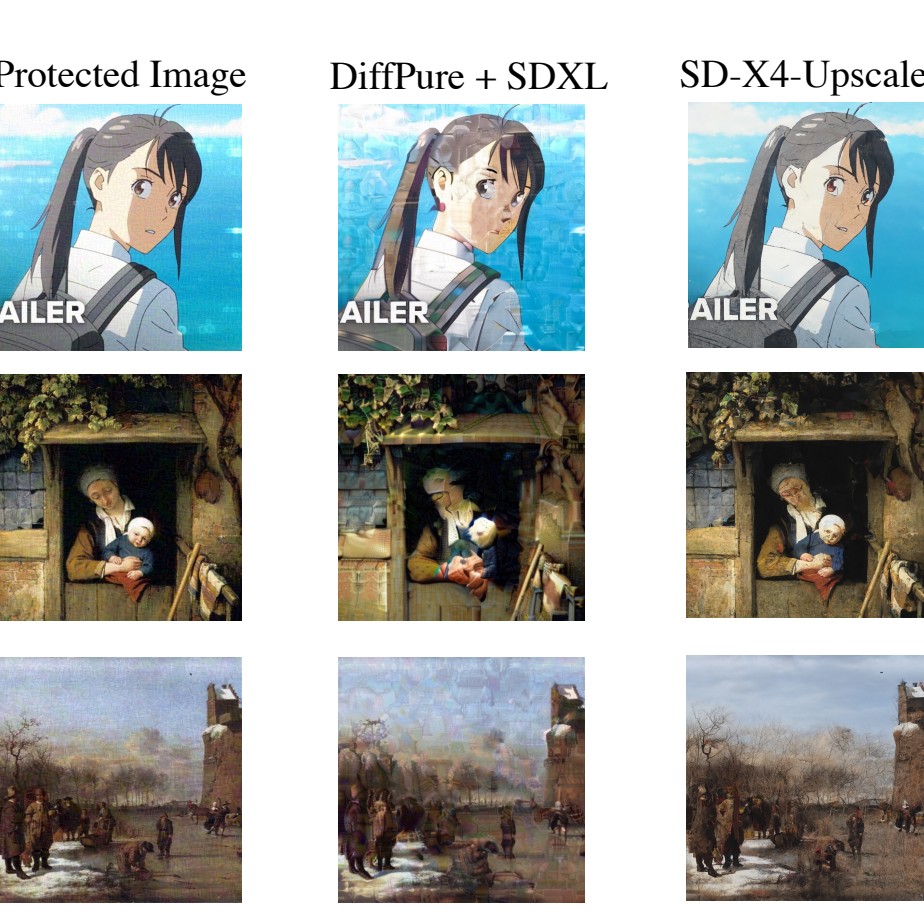

Figure 18: **LDM as Purifier**: When protection is applied to the given LDM, DiffPure combined with the LDM will fail to function effectively, as the purification process can be easily fooled. Additionally, the LDM-based upscaler lacks stability, often resulting in poor detail quality.

