# OpenReview forum: "Rethinking Adversarial Attacks as Protection Against Diffusion-based Mimicry"
_ICLR.cc/2025/Conference — Submitted to ICLR 2025_

### Official Review · Reviewer_6bSH · 2024-10-28

**Soundness:** 3
**Presentation:** 3
**Contribution:** 3
**Rating:** 8
**Confidence:** 4

**Summary:**

This paper investigates the protective protection against diffusion-driven editting/mimicrying. The protective protection in previous works is generally adversarial perturbations designed against latent diffusion models due to the huge impact of Stable Diffusion model families. Following the findings of previous literature that discovered current perturbations mainly attacks the VAE part of the LDM, this paper conducts experiments on attacking PDMs and find they are (empirically) surprisingly more robust than LDMs, even under EOT-based adaptive attacks and recently proposed adaptive attacks. Based on this understanding, the paper proposes a simple yet effective defense (or attack against defensive protection) named PDM-Pure, which directly uses pre-trained PDMs and SDEdit to purify a given image and remove the injected noises. The results on several SOTA diffusion models and attacks demonstrates the effectiveness of PDM-Pure.

**Strengths:**

- The motivation for testing protective perturbations designed for LDMs on PDMs is interesting and novel. This paper's discoveries and results are also fresh and insightful.

- The preliminary results challenge the common belief that DMs are highly susceptible to adversarial attacks, offering new perspectives for the community.

- This paper tries to evaluate PDMs against adaptive attacks and the attack from the latest work (Shih et al., 2024), which should be encouraged.

- The paper is generally well-written and motivated. The proposed method is straightforward, effective, cleverly designed, and clearly explained.

- The experiments cover a wide range of cutting-edge diffusion models, with impressive results that likely open new avenues for disruption-based protection against diffusion-driven editing and mimicry.

- The authors acknowledged the trade-off between purification effectiveness and image quality, which is commendable.

**Weaknesses:**

- While I appreciate the effort to evaluate PDMs against adaptive attacks and the recent attack proposed by Shih et al. (2024), the results in Figures 3 and 4 seem to be a rudimentary case study rather than a comprehensive experiment. Thus, the claim that PDMs are also robust against adaptive attacks is not very convincing. Moreover, the claim in Fig. 4 that Shih et al.'s attack needs perturbation buget of $\ell_{\infty}>150/255$ gives a sense that this attack is not truely adaptive or not correctly applied, because at the perturbation budget of this level, any image can be converted into a gray image (i.e., (127,127,127), or #7f7f7f in Hex, at every pixel), making it impossible for the model to identify original features. Besides, it would be helpful for the authors to clarify the experimental setup used in these figures (i.e., directly use SDEdit to noise the protected image and denoise it, if I understand correctly). Because at first glance, the images appear to be merely restored rather than edited (e.g., in typical image editing, one might expect transformations like a sunset being altered to a sunrise).

- Although the authors provide a reasonable explanation for why PDMs seem more robust than LDMs, it remains unclear why PDMs demonstrate such strong empirical robustness. Since PDMs are still neural networks, one might expect them to also be vulnerable to adversarial attacks. I recommend that the authors further analyze and hypothesize the reasons behind this robustness to enhance the reliability of their findings and deepen our understanding of the method’s mechanisms. More interestingly, recent literature have discovered the potential of diffusion models to be applied for classification tasks [1]. Does that mean we can obtain a zero-shot and adversarially robust classifier for free? I encourage the authors to discuss this possibility and any challenges they foresee.

- From a technical perspective, the contributions in this work seem similar to those in GrIDPure and DiffPure (specifically similar to DiffPure, whose idea is also use noising-denoising nature of DMs to purify adversarial noises). A more detailed explanation and comparison would help to clarify the paper’s uniqueness and value.

The reviewer would like to actively participate in the disucssion and will adjust the rating accordingly.

Minor:

- Figure 3: PDM should robustness -> PDM shows robustness
- L252-257: please consider unifying the abbreviation of EOT
- L267: as shown in the last block of Figure 4) -> as shown in the last block of Figure 4
- L296: it barely work -> it barely works

Ref:

[1]: Li et al. Your diffusion model is secretly a zero-shot classifier. ICCV 2023.

**Questions:**

- How does the technical similarity between your paper and prior works such as GrIDPure and DiffPure impact the novelty and contribution of your study?

- PDMs are also neural networks after all. What might account for their empirical robustness to adversarial perturbations?

- Can the adversarial robustness of PDMs in generative tasks also be applied to classification tasks? I recently came across studies by Chen et al. [2,3] (I did not go into the details of these papers as they are full of theories and mathematical proofs), which propose that diffusion models can serve as certifiably robust classifiers, but I'm not sure whether this is strongly connected to your discovery on PDMs. Could the authors share their perspective on this potential crossover?

- I think another limitaion is that PDMs may not scale as good as LDMs, making the potential of PDM-Pure for large scale purification difficult. Can the authors share their opions and comments on this aspect?


Ref:

[2]: Chen et al. Robust Classification via a Single Diffusion Model. ICML 2024.

[3]: Chen et al. Diffusion Models are Certifiably Robust Classifiers. NeurIPS 2024.

---

> ### Author Response · Authors · 2024-11-17
> **Thanks for your very careful review and valuable comments!**
>
> We express our gratitude to the reviewer for the careful review and valuable feedback. Here I will answer some questions that can hopefully address your concern:
>
> > W1: (1) Figures 3 and 4 seem to be a rudimentary case study rather than a comprehensive experiment; (2) Fig. 4 that Shih et al.'s attack needs perturbation buget of gives a sense that this attack is not truely adaptive or not correctly applied (3) settings of SDEdit
>
> (1) Thank you for pointing it out! We will include additional results in the appendix in the next version.
>
> (2) Regarding the settings, we follow those in Shih et al. Their approach maintains a budget in the encoded latent space using LPIPS, as opposed to relying on an L-infinity budget in pixel space.
>
> (3) For SDEdit, we use step sizes smaller than 0.5, specifically 0.1, 0.3, and 0.5. Since no text prompts are used as conditions, the resulting changes are limited. However, this does not impact performance when text prompts are applied, as evidenced by image inpainting examples in prior protection papers, such as Mist.
>
>
> > W2: (1) I recommend that the authors further analyze and hypothesize the reasons behind this robustness to enhance the reliability of their findings and deepen our understanding of the method’s mechanisms. (2) recent works about diffusion model as a zero-shot classifier
>
> (1) Thanks for pointing it out. Regarding why PDMs are so robust, while the rigorous theory of the mechanism remains largely unexplored, we attempt to explain it intuitively here: (*) From the training of the network: The denoisers in PDMs are trained with extensive data augmentation, including Gaussian noise, which makes them quite robust to small perturbations; while for LDMs, the networks are fooled because the inputs are already out of the distribution of the dataset due to the huge perturbation in the latent space (**) From the structure of diffusion models: The input to the network includes a high level of randomness, as new noise is injected at different timesteps, making the whole system includes high level of randomness, making it robust.
>
> (2) Yes, conditional diffusion models can indeed function as zero-shot classifiers [1]. Furthermore, studies such as [2, 3] highlight that diffusion classifiers exhibit strong certified robustness. These works strongly support the insights presented in our paper. Due to the inherent stochasticity of the diffusion process, it is notably challenging to design adversarial attacks against diffusion models, unlike the more straightforward vulnerabilities observed in other deep neural networks.
>
> > Comparison with Diff-Pure and GrIDPure
>
> The focus of our study is the adversarial-robustness of PDMs, which is novel and insightful. PDM-Pure is motivated from this finding, while GrIDPure does no realize that it is because of the PDM that matters (they did not mention any PDM in their paper). As Line 356-358 we propose a generalized framework to purify any noise, which is based on the assumption that PDM is robust to any small noise since it is adversarially robust.
>
>
> Both PDM-Pure and Diff-Pure operate on the assumption that diffusion models can eliminate out-of-distribution noise patterns through a diffuse-then-denoise process. However, our motivation diverges: given the insight that PDMs are exceptionally robust to adversarial attacks, we hypothesize that they can mitigate "any" noise using the Diff-Pure pipeline, provided it’s sufficiently robust. LDMs, by contrast, cannot achieve this due to their vulnerability to adversarially crafted noise. Notably, Diff-Pure did not highlight the adversarial robustness of PDMs, which is a core focus of our paper.
>
>
>
> > Q1: How does the technical similarity between your paper and prior works such as GrIDPure and DiffPure impact the novelty and contribution of your study?
>
> See W3
>
> > Q2: PDMs are also neural networks after all. What might account for their empirical robustness to adversarial perturbations?
>
> See W2
>
> > Q3:  Can the adversarial robustness of PDMs in generative tasks also be applied to classification tasks?
>
> Yes, [2,3] exactly support the findings in our paper. It is promising for future works to design classification / regression models based on diffusion model, which shows strong adversarial robustness.
>
> > Q4: I think another limitaion is that PDMs may not scale as good as LDMs, making the potential of PDM-Pure for large scale purification difficult. Can the authors share their opions and comments on this aspect?
>
> That is not true. First, PDMs are very strong—models like DeepFloyd, Hourglass Diffusion Transformers, and Ediff-I demonstrate this strength. Additionally, very recent work like Edify-Image [a] pushes the scale to the extreme. By applying techniques like pyramid Laplacian structures, denoising becomes more effective. I believe it will be quite promising to use large PDMs as purifiers.
>
> [a] https://huggingface.co/papers/2411.07126

---

> > ### Comment · Reviewer_6bSH · 2024-11-20
> > **Thank you for your response.**
> >
> > Dear authors, thank you for the response. Sorry for my late reply due to my deadlines. I have also read other reviews and your rebuttal. Below are my further comments:
> >
> > W1: Thank you for the response. It will be good if you include additional, formal results in the new version. Besides, I understand that you experiment follows the attack of Shih et al., my point is that claiming this attack to be "adaptive" and illustrating its budget of >150 in the main paper is not appropiate, because this attack is even no more "adaptive" than a simple gray-scale change under the same budget. So please consider fixing it. For the settings of SDEdit, thanks for the clarification and I understand your performance is not affected by the prompts. Please add these details into your next version.
> >
> > W2: Thanks for the response. I personally agree with your point that the Gaussian noising process of DM training somewhat serves as a data augmentation similar to adversarial training, making PDMs robust. Actually, the LDMs are also robust in the latent space, but the problem is the weakness of VAEs, which can bring very large perturbation at the latent space with small changes in pixel space. Please consider adding more discussions (including the details of classifier part, I think these works are somewhat interconnected) into your next version, which would be insighful for your readers.
> >
> > W3 & Q1 & Q3: Thanks for the response. Overall they addressed my concerns. For Q2, please see my comments above.
> >
> > Q4: Thanks for the response. While I appreciate the authors' effort on bringing state-of-the-art pixel space diffusion models into my vision, I somewhat keep skeptical about these scaling-up works (e.g., Edify Image). Typically, PDMs are not easy to train, so many works are intended to scale down the resolution of diffusion process and then scale the image up to the pixel space. LDM actually can be classified as a kind of this line, which uses VAEs to scale down the resolution. Similarly, Imagen uses a super-resolution pipeline to scale down the diffusion process and scale up the image after diffusion. So it is actually unclear whether these scaling-down process will also introduce new security vunlerbilities (e.g., like adversarial perturbations for VAEs). Therefore, I remain skeptical about whether these "scaled-up PDMs" still works as good as small scale PDMs that are directly trained in the pixel spaces.
> >
> > Overall I believe it is a good work and I lean to the acceptance side. If my concerns above are well addressed I'll be raising my score to 8 and try to convince other reviewers. Otherwise I will keep my score.
> >
> > Minor:
> >
> > Line 472, please fix "Figure ??".

---

> ### Author Response · Authors · 2024-11-20
> **Thanks for your response.**
>
> Thanks for your response and again for your careful review! I am happy to answer your following questions and hopefully it can address your concerns.
>
> > W1
>
> Yes, I believe that your thoughts align exactly with what we want to convey in the paper! We also think that the attacks presented in Shih et al. are not in a good setting. However, Shih et al. is a recent work that claims to have found a good adaptive attack for PDM. We applied their attack and found that it is not in a good setting: the budget norm is too large, which actually introduces significant perturbations into the image.
>
> We will add the clarification about SDEdit settings in the next revision in a few days.
>
> > W2
>
> Thanks, we will add more discussions about that in the next revsion.
>
> > Q4
>
> Yes, I agree that scaling up PDM will be more challenging compared to LDM, many tricks should be applied otherwise the training will be too costful. However, for PDM-Pure, we may not require an expert-level PDM scaled to extensive levels. Since we're not generating images from pure noise but instead using SDEdit, the current PDMs, such as Edify-Image and DeepFloyd, might already be sufficiently strong.

---

> ### Comment · Reviewer_6bSH · 2024-11-20
> **I have raised the score**
>
> Dear authors, thanks for the new responses. Accordingly, I have raised my score to an 8. I think this paper brings some novel insights and proposes a simple yet practical method, which is an important contribution to the field and thus worth acceptance at this year's ICLR. I also agree with other reviewers that some related works about using DMs to purify adversarial noises, as well as more discussions, should be added to the next version of the paper. This paper is likely to have high impact beyond the current scope and I hope the authors to extend the discussion of this paper for future work.

---

> > ### Author Response · Authors · 2024-11-20
> > **Thanks for your acknowledgement**
> >
> > We are happy to see that we have addressed your concern. Thanks for your acknowledgement of our paper, we will include the discussions during the rebuttal session into our next revision.

---

### Official Review · Reviewer_GDQ5 · 2024-11-02

**Soundness:** 3
**Presentation:** 3
**Contribution:** 3
**Rating:** 8
**Confidence:** 3

**Summary:**

The paper explores the robustness of pixel-space diffusion models (PDMs) against adversarial attacks, contrasting them with latent diffusion models (LDMs). The authors highlight a significant gap in existing literature, which predominantly focuses on LDMs while neglecting PDMs. Through extensive experimentation, they demonstrate that traditional adversarial attack methods fail to effectively compromise PDMs, suggesting that these models possess a higher degree of robustness. The authors propose a new framework, PDM-Pure, which utilizes PDMs as a universal purifier to eliminate adversarial perturbations generated by LDMs, thereby enhancing image integrity.
In conclusion, the paper not only fills a critical gap in the literature but also provides actionable insights that could influence future research and applications in the field. Its novel contributions and practical implications make it a valuable addition to the conference proceedings.

**Strengths:**

- Novel Contribution: The paper identifies a critical oversight in the current understanding of adversarial attacks on diffusion models by focusing on PDMs, which have been largely neglected in prior research.

- Robust Experimental Validation: The authors conduct comprehensive experiments across various architectures and datasets, demonstrating the superior robustness of PDMs compared to LDMs against adversarial perturbations.

- Practical Implications: The introduction of the PDM-Pure framework offers a practical solution for purifying images from adversarial attacks, which could have significant implications for protecting intellectual property in digital media.

- Clear Presentation: The paper is well-structured and clearly presents its findings, making it accessible to readers with varying levels of expertise in the field.

**Weaknesses:**

- Limited Scope of Experiments: While the experiments are extensive, they may not cover all potential adversarial attack methods or variations in model architectures that could further validate the findings.

- Assumption of Robustness: The conclusion that PDMs are universally robust may require further exploration under different conditions or with more sophisticated attack strategies that were not tested in this study.

- Lack of Discussion on Limitations: The paper could benefit from a more thorough discussion of the limitations of their approach and potential scenarios where PDMs might still be vulnerable to attacks.

**Questions:**

Would you also add a code to this framework, please?

---

> ### Author Response · Authors · 2024-11-17
> **Thanks for your review!**
>
> We express our gratitude to the reviewer for acknowledging our paper and providing valuable feedback. Here I will answer some questions that can hopefully address your concern:
>
> > W1: While the experiments are extensive, they may not cover all potential adversarial attack methods or variations in model architectures that could further validate the findings.
>
> We show the **empirical** adversarial robustness of PDMs by applying various attacks as far as we know, we conclude that it is clear that PDM is much more robustness than LDMs. For model archs, U-Net and Transformers are two main structures for diffusion models and are used by most models, while there is currently not transformer for PDMs, we believe there could shares similar robustness.
>
> > W2: The conclusion that PDMs are universally robust may require further exploration under different conditions or with more sophisticated attack strategies that were not tested in this study.
>
> We tested various of attacks including latent attack and EOT, targeted attack and un-targeted attack, PDM shows strong empirical adv-robustness. Also, there are some recent evidences that diffusion model can be strong classifier with certified robustness [a, b].
>
> > W3: The paper could benefit from a more thorough discussion of the limitations of their approach and potential scenarios where PDMs might still be vulnerable to attacks.
>
> Thanks for pointing it out. Currently we do not see an attack that can effectively attack PDMs. We will have a discussion of it in our final version.
>
>
> [a]: Chen et al. Robust Classification via a Single Diffusion Model. ICML 2024.
>
> [b]: Chen et al. Diffusion Models are Certifiably Robust Classifiers. NeurIPS 2024.

---

> ### Comment · Reviewer_GDQ5 · 2024-11-21
> **future work and code base**
>
> Thanks, for answering most of my questions.
>
> ad W2: I would not give so much on EoT actually (even though is important to have some evaluations on it), this just means that the attack can be successful after certain transformation, too.
> Honestly, I recommend study semantic adversarial attacks (not gradient-based attacks) - mostly created by diffusion models [1,2]:
>
> - [1] https://github.com/WindVChen/DiffAttack
> - [2] https://arxiv.org/abs/2309.07398
>
> I hope this strengthens your work.
>
> ad W3: like W2
>
> Q1: Would you share a code base? I don't understand, why not sharing. An initial partial code base would be good. People would be motivated to compare and consequently cite more of your paper.
> You also emphasize that your paper is empirical work.

---

> > ### Author Response · Authors · 2024-11-21
> > **Thanks for replying**
> >
> > Thanks for your reply!
> >
> > > About EoT
> >
> > Here we apply EoT to aggregate random gradients (because the input of diffusion model is noised), not to make it successful for certain transformations.
> >
> > > Semantic Attacks
> >
> > Actually (Shih et al., 2024) is one kind of semantic attacks, it does not work well actually. Also, semantic attacks tend to have large perturbation, which may sacrifice the quality of personal image when protecting it.
> >
> > > Code-base
> >
> > Sure, we are happy to share the codebase on Github once the paper's anonymization is removed.

---

### Official Review · Reviewer_sKMW · 2024-11-02

**Soundness:** 3
**Presentation:** 2
**Contribution:** 2
**Rating:** 3
**Confidence:** 4

**Summary:**

This paper primarily focuses on adversarial attacks targeting diffusion models. The authors first discover that previously effective attacks on latent diffusion models (LDMs) are no longer successful against pixel-based diffusion models (PDMs). Furthermore, the authors demonstrate that PDMs can serve as an effective off-the-shelf purifier, capable of eliminating adversarial patterns generated by LDMs. This preservation of image integrity effectively removes protective perturbations of various scales encountered in applications like Glaze and Mist.

**Strengths:**

This paper focuses on an intriguing topic regarding attackers who aim to target pixel-based diffusion models and attempt to eliminate protective perturbations. It highlights that tools relying on adversarial perturbations cannot consistently safeguard artists from the misuse of generative AI.

Additionally, the authors conducted several adaptive attacks, transitioning adversarial attacks from Latent Diffusion Models (LDMs) to Pixel Diffusion Models (PDMs). They also introduce their straightforward yet effective method, PDM-PURE.


They also conducted various experiments on advanced protection methods, such as Photoguard, Glaze, and Mist, which indicate that these methods cannot effectively prevent the misuse of generative AI.

**Weaknesses:**

To enhance the overall interest of the story, I believe addressing the following points could improve this work:

1. The paper is not well-prepared, with many typos, formatting errors, and grammar issues:

   - In lines 15 and 436, there's an extra space before the comma.
   - In line 19, "PDM" is not well-defined initially, yet it is repeatedly redefined in several places, such as in lines 163 and 477.
   - There are typos in lines 266 and 267: "to" and ")".
   - In line 265, where it says `$l_\infty > 150/255$`, do you mean $\delta$?
   - Typo in line 281: "should."
   - Typo in line 295: "decode."
   - In line 469, `t*`; in line 471, `??`.


2. Some techniques are not well-explained:
   - While I understand that attacks designed for latent space are less effective in pixel space, the authors seem to spend excessive space discussing this point (Section 4 already addressed this point, but Section 6.2 reiterates it.). It’s fairly intuitive that different optimization losses lead to this limitation. What would be more insightful is a deeper exploration of adaptive attacks in pixel space. Here are some specific questions:

     1. Could you elaborate on your two categories of attacks (lines 247-248)? From my understanding, a reasonable approach might be to adapt the loss in Equation 1 to match the loss function used in pixel-based models. Additionally, it could help to incorporate information from intermediate layers, run EOT multiple times, and use these gradients to update adversarial images. Based on the current description, it’s unclear if the adaptive attack implemented here is sufficiently strong.
      2. The authors use Figure 2 to explain why pixel space adversarial attacks are not effective, particularly due to the large overlap. However, for traditional adversarial attacks on ResNet models using the ImageNet dataset, the adversarial examples are generated through pixel-level updates, no? This method has also proven to be very successful. any difference here?
     3. In line 318, what do you mean by "domain shift"?  and why?

   - The explanation of PDM-Pure is also insufficiently detailed (taking up less than half a page on page 7). Specifically:

     1. In line 362, the authors state that "the adversarial pattern should be removed," but the rationale behind this is unclear.
     2. In line 465, it’s mentioned that "LDMs cannot be used to purify the protected images." This seems like an overstatement; as far as I know, LDM (Stable Diffusion) has been used in prior work[1] to successfully attack Glaze.


[1] Adversarial Perturbations Cannot Reliably Protect Artists From Generative AI. arxiv: 2406.12027

**Questions:**

Overall, the research topic and experiments are good. I would suggest that the authors fix all typos and  focus more on (1) demonstrating convincingly that adaptive attacks are ineffective, and (2) discussing why PDM-Pure performs well.
Additionally, there’s no need to spend excessive space explaining why adversarial attacks designed for LDMs are less effective for PDMs, as this is fairly intuitive.

---

> ### Author Response · Authors · 2024-11-17
> **Thanks for your very careful review!  Part 1/2**
>
> Thanks for your very careful review! Here I will answer some questions that can hopefully address your concern:
>
> > W1: The paper is not well-prepared, with many typos, formatting errors, and grammar issues:
>
> Sorry for that, we changed all the typos that have been mentioned, please refer to ** Revision Summary ** for the updates.
>
> > W2: While I understand that attacks designed for latent space are less effective in pixel space, the authors seem to spend excessive space discussing this point (Section 4 already addressed this point, but Section 6.2 reiterates it.). It’s fairly intuitive that different optimization losses lead to this limitation.
>
> Section 4 aims to demonstrate the remarkable adversarial robustness of the denoising-diffusion process. In Section 4.1, we present attack results using the diffusion loss, a widely used method to attack diffusion models. Section 4.2 details our efforts to craft more adaptive attacks specifically for PDMs, while Section 4.3 explains why LDMs are vulnerable despite being diffusion models.
>
> Section 6.2, part of the experiments section, provides a detailed analysis of the settings and metrics used in Sections 4.2 and 4.3. Although there may be some overlap, we will reorganize these sections for clarity in the final revision.
>
> Even when using the same optimization loss (Eq. 1) — the well-known diffusion loss — we find that we can effectively attack LDMs, but not PDMs. This is a significant observation, as we are the first to highlight this difference, which has been largely overlooked in this field.
>
>
> > W 2.1: Could you elaborate on your two categories of attacks (lines 247-248)? From my understanding, a reasonable approach might be to adapt the loss in Equation 1 to match the loss function used in pixel-based models. Additionally, it could help to incorporate information from intermediate layers, run EOT multiple times, and use these gradients to update adversarial images. Based on the current description, it’s unclear if the adaptive attack implemented here is sufficiently strong.
>
> Sure, we are glad to elaborate our two kinds of attacks (lines 247 - 248):
>
> - Attack the full pipelines, which is adopted in [a], regard the multi-step diffusion-based editing process of a clean images as a end-to-end pipeline, which is typically impractical but turns out to be stronger in [a].
>
> - Attack the diffusion loss means attack the PDM follow Equation 1, but changing $z_t$ to $x_t$
>
>
> We indeed proposes to incorporate information from middle layers of U-Net and run EOT to aggregate gradients in the same section (line 253-257). None of the proposed attacks can effectively attack PDMs.
>
> > W 2.2: The authors use Figure 2 to explain why pixel space adversarial attacks are not effective, particularly due to the large overlap. However, for traditional adversarial attacks on ResNet models using the ImageNet dataset, the adversarial examples are generated through pixel-level updates, no? This method has also proven to be very successful. any difference here?
>
> That’s a good question. The difference is that, before input into the network, a large scale of noise is added to $x$, the input is actually sampled from a distribution $N(ax, b^2I)$ where $a,b$ are defined by the diffusion process. While for previous adversarial attacks, the pixel updates will be directly input into the classifier, making the attack much more effective.
>
>
> > W 2.3: In line 318, what do you mean by "domain shift"? and why?
>
> Domain-shift means: the attacked image results in a latent $z’_t$ that significantly deviates from the training / testing distribution of $z_t$ (clean latents). The difference of it from adversarial perturbation is that the perturbation of $z_t$ is much larger, Figure 2 in [b] gives a good visualization for that.
>
>
>
>
> [a] H. Salman, A. Khaddaj, G. Leclerc, A. Ilyas, and A. Madry. Raising the cost of malicious ai-powered image editing. arXiv preprint arXiv:2302.06588, 2023.
>
> [b] Xue, H., Liang, C., Wu, X., & Chen, Y. (2023, October). Toward effective protection against diffusion-based mimicry through score distillation. In The Twelfth International Conference on Learning Representations.
>
> [c] Adversarial Perturbations Cannot Reliably Protect Artists From Generative AI. arxiv: 2406.12027

---

> > ### Author Response · Authors · 2024-11-17
> > **Thanks for your very careful review! Part 2/2**
> >
> > > W3.1: In line 362, the authors state that "the adversarial pattern should be removed," but the rationale behind this is unclear.
> >
> >
> > This assumption is commonly adopted in other studies, such as Diff-Pure [c], where diffusion models are utilized to remove adversarial noise from classifiers. The underlying rationale is that the added Gaussian noise diffuses the adversarial pattern, and the subsequent denoising process effectively mitigates the noise. However, our motivation diverges slightly: the original Diff-Pure does not account for adversarial noise targeting the diffusion model itself. If such noise successfully compromises the diffusion model, Diff-Pure would undoubtedly fail. Nevertheless, based on our findings that PDM is highly robust to adversarial noise, we propose using PDM in combination with Diff-Pure as a universal purifier.
> >
> > > W3.2: In line 465, it’s mentioned that "LDMs cannot be used to purify the protected images." This seems like an overstatement; as far as I know, LDM (Stable Diffusion) has been used in prior work[1] to successfully attack Glaze.
> >
> > First, based on our experience, Glaze is not an effective adversarial attack against diffusion models because it does not target the diffusion loss directly. Instead, it focuses more on style transfer, which limits its applicability. Notably, it even fails to perform effectively in SDEdit during our main experiments.
> >
> > On the other hand, [1] utilizes a diffusion-based upsampler with an architecture distinct from the original diffusion model. The upsampler fine-tunes an image encoder to generate conditions for the diffusion process, then samples from pure Gaussian noise. This highlights a critical difference: the conditional image encoder in the upsampler differs significantly from the latent encoder used in standard diffusion models. Consequently, if adversarial noise is crafted to target the conditional encoder, [1] may fail. PDM-Pure instead is a more robust way to do purification.

---

> > ### Comment · Reviewer_sKMW · 2024-11-20
> > **reply**
> >
> > Thanks for the reply!
> >
> > > We changed all the typos mentioned. Please refer for the updates.
> >
> > I can still find some typos in L469 and L473, e.g., ?
> >
> > > None of the proposed attacks can effectively attack PDMs.
> >
> > Good, please add these results somewhere.
> >
> > > we find that we can effectively attack LDMs, but not PDMs. This is a significant observation, as we are the first to highlight this difference, which has been largely overlooked in this field.
> >
> > I think that's why people do adaptive attacks first for different methods.
> >
> > > "LDMs cannot be used to purify the protected images."
> > I am still not convinced by this statement.
> >
> > Overall, the current draft is not well-prepared and a bit messy. The authors dedicate excessive space to explaining why adversarial attacks designed for LDMs are less effective on PDMs (I don't see why this is challenging), while offering limited focus on why the proposed method can work (feel like not much special things to show?).
> > In particular, not clear what's the key insights and takeaways for researchers to get from these results.

---

> > > ### Author Response · Authors · 2024-11-22
> > > **Thanks for your reply**
> > >
> > > Thanks for your reply.
> > >
> > > > About typos
> > >
> > > Revision-v2 will be the version after proof-reading.
> > >
> > > > None of the proposed attacks can effectively attack PDMs.
> > >
> > > We put more samples in the Appendix Figure 17. Please check Revision Summary v2 to check the updates.
> > >
> > > > "LDMs cannot be used to purify the protected images." I am still not convinced by this statement.
> > >
> > > For most image-to-image settings, if the noise is generated for Stable Diffusion (SD), popular models like SD-1.5, SD-2.1, and SDXL cannot be effectively used as purifiers. This limitation arises because these models are easily fooled when employing methods like DiffPure. Reference [1] focuses on weak protection mechanisms such as Glaze, which fail to provide adequate protection in image-to-image settings. As demonstrated in Appendix Figure 18, protection generated for SD-1.5 is capable of fooling SDXL during the DiffPure process.
> > >
> > > Moreover, [1] proposes using the upscaler SD-X4 to remove adversarial perturbations. It is important to note that SD-X4 operates fundamentally differently from DiffPure. Unlike DiffPure, SD-X4 uses the raw pixel input of a low-resolution image as a condition and starts denoising from pure Gaussian noise, making it incomparable with our PDM-Pure. Our experiments reveal that SD-X4 is highly unstable, leading to significant loss of detail, as shown in Figure 18.
> > >
> > > In contrast, PDM stands out as a superior and more universal solution. To date, no noise has been found that can also fool PDM-Pure, making it a more reliable option.
> > >
> > > > Overall, the current draft is not well-prepared and a bit messy. The authors dedicate excessive space to explaining why adversarial attacks designed for LDMs are less effective on PDMs (I don't see why this is challenging), while offering limited focus on why the proposed method can work (feel like not much special things to show?). In particular, not clear what's the key insights and takeaways for researchers to get from these results.
> > >
> > > First, the loss function designed for LDM (Eq. 1) is not exclusive to LDMs but is, in fact, the standard training loss for all diffusion models. We are the first to challenge and correct the misconception that attacking the diffusion loss will effectively fool all diffusion models. This vulnerability is specific to LDMs and does not extend to PDMs. On the contrary, the diffusion process itself, particularly in PDMs, is quite robust.
> > >
> > > This insight is not obvious, and, in fact, no one in the field has given it proper consideration. The research focus has predominantly been on developing new methods to make the protections for LDMs faster or perform better, while **PDMs have been largely overlooked**. Our findings are critically important for this area, more works should be done to protect our images from strong PDMs (e.g. recent Edify-Image [2]).
> > >
> > >
> > >
> > >
> > >
> > >
> > >
> > >
> > >
> > >
> > >
> > >
> > >
> > >
> > >
> > >
> > >
> > > Key insights and takeaway:
> > >
> > >  (1) Latent Diffusion Models (LDMs) are significantly more vulnerable compared to Pixel Diffusion Models (PDMs). Previous attacks have created a misleading perception that diffusion models are inherently easy to attack, suggesting that personal images can be protected simply by adding adversarial perturbations. However, this assumption does not hold for PDMs.
> > >
> > > (2) The diffusion process, particularly in PDMs, is far more robust than initially assumed. Without relying on encoder-decoder architectures, PDMs demonstrate robustness against all adaptive attacks. This robustness make PDMs universal purifiers, as long as the model is strong enough.
> > >
> > >
> > > [a] https://huggingface.co/papers/2411.07126

---

> > > > ### Comment · Reviewer_sKMW · 2024-11-27
> > > > **reply**
> > > >
> > > > Thanks for your reply!
> > > >
> > > > > Key insights and takeaway:
> > > >
> > > > I am not entirely convinced by the claim that "PDMs demonstrate robustness against all adaptive attacks." Considering the successful attack on LDMs, it seems intuitive that a carefully designed adaptive attack tailored for PDMs could potentially be effective. Since there is no certified robustness, I am concerned that someone might develop a powerful adaptive attack and demonstrate that it can indeed compromise your method.
> > > >
> > > > I’d be careful with this claim. It would be good to add more discussion about why there won’t be a successful attack on PDM. Thus, I'd like keep my current score.

---

> > > > > ### Author Response · Authors · 2024-11-28
> > > > > **Thanks for your response and we are always open to further questions**
> > > > >
> > > > > Thanks for your reply.
> > > > >
> > > > > > I am not entirely convinced by the claim that "PDMs demonstrate robustness against all adaptive attacks.
> > > > >
> > > > > Our intention was to convey that all the existing adaptive attacks we attempted failed to fool PDMs. PDMs show strong empirical robustness and are worth attention and investigation from the community.
> > > > >
> > > > > > Considering the successful attack on LDMs, It seems intuitive that a carefully designed adaptive attack tailored for PDMs could potentially be effective
> > > > >
> > > > > We respectfully disagree. As discussed in Section 4.3 of our paper, the successful attacks on LDMs are largely due to their **unique** encoder-decoder structure (not very related to the diffusion process) [1]. The input to the denoiser in a diffusion model is already highly chaotic (with significantly larger perturbations). Allowing such perturbations in pixel space for PDMs is not feasible. This highlights that the mechanism for attacking LDMs is fundamentally different from attacking diffusion models in general, and the success of attacks on LDMs is not generalizable to general DMs.
> > > > >
> > > > >
> > > > > > Since there is no certified robustness, I am concerned that someone might develop a powerful adaptive attack and demonstrate that it can indeed compromise your method.
> > > > >
> > > > > In most cases, certified robustness guarantees are limited to a small scope. We appreciate Reviewer 6bSH for highlighting papers [2,3] that demonstrate strong certified robustness of diffusion models (specifically PDMs) in classification tasks. The underlying reason for this robustness aligns with the explanations we provided in the following section:
> > > > >
> > > > >
> > > > >
> > > > > > I’d be careful with this claim. It would be good to add more discussion about why there won’t be a successful attack on PDM.
> > > > >
> > > > >
> > > > > We can provide some discussion here about why PDM could be adversarially robust (*) From the training of the network: The denoisers in PDMs are trained with extensive data augmentation, including Gaussian noise, which makes them quite robust to small perturbations; while for LDMs, the networks are fooled because the inputs are already out of the distribution of the dataset due to the huge perturbation in the latent space (**) From the structure of diffusion models: The input to the network includes a high level of randomness, as new noise is injected at different timesteps, making the whole system includes high level of randomness, making it robust.
> > > > >
> > > > >
> > > > >
> > > > > [1]: Xue et al. Toward effective protection against diffusion-based mimicry through score distillation. ICLR 2024
> > > > >
> > > > > [2]: Chen et al. Robust Classification via a Single Diffusion Model. ICML 2024.
> > > > >
> > > > > [3]: Chen et al. Diffusion Models are Certifiably Robust Classifiers. NeurIPS 2024.

---

> ### Comment · Reviewer_6bSH · 2024-11-29
> **A warm discussion from Reviewer 6bSH.**
>
> Hi dear colleague, thank you very much for your valuable time and thoughtful review. I have carefully read your comments and the subsequent discussion with the authors. I completely agree with your point regarding the authors' initial claim that "PDMs are robust against adaptive attacks," which indeed lacks sufficient theoretical support and could potentially mislead readers. I also share similar concerns about the problematic claim that PDMs are robust against Shih et al.'s attack for budget 150/255 in my initial comments (as noted in my W#1). They have since acknowledged these concerns and have worked on revising their claims accordingly.
>
> That being said, I would like to respectively emphasize that while I agree that PDMPure may eventually be defeated by stronger adaptive attacks in the future, I humbly don't believe this should be the sole reason for rejecting the paper. After all, security is an evolving game between attacks and defenses, the community's understanding goes step-by-step, so it is natural for defenses to be challenged over time, especially for attacks/defenses against diffusion models, a relatively new research field. It is not fresh to see that strong empirical defenses published on top conferences are quickly defeated in a few months (or even days). However, this should not diminish the value of such works. Overall, this paper offers valuable insights to the community and presents a defense that, despite its simplicity, effectively defeats many existing complex defenses.
>
> In light of this, even though the paper's claims have room for improvement and there is no theoretical guarantee for its robustness, I believe it should not be dismissed entirely. The paper is well-written, contributes new perspectives to the field, and introduces a practical and effective method that is worth sharing with the community. Overall, this paper does not contain obvious mistakes or evaluation faults (at least in the current manuscript, to my knowledge), and I believe it would be unfair for the authors to reject it simply based on the reason that "it may be defeated by future stronger attacks".
>
> Would you mind reconsidering the rating for this paper, taking these points into account? Thank you again for your time and thoughtful feedback.

---

> ### Comment · Reviewer_sKMW · 2024-11-29
> **reply to the authors and Reviewer 6bSH**
>
> Thank you for the further discussion! I'm pleased to see such a heated discussion about this paper! Regarding my concerns:
>
> 1. "The paper is well-written, contributes new perspectives to the field, and introduces a practical and effective method that is worth sharing with the community. "
>
> I do not fully agree with the submitted version. There are numerous typos, and the formatting is not well done. Even after the authors revised it twice, I still have concerns about the writing. The current narrative focuses too heavily on demonstrating that "attacks for LDM do not work for PDM" and emphasizes the need to focus on PDM. While this reasoning is intuitive, it does not require extensive elaboration. Meanwhile, the authors have devoted less effort to convincingly explaining why PDM-PURE is strong, particularly in Section 5.
>
> I would suggest the following: (1) Invest more effort into understanding adaptive attacks to see if PDM is genuinely robust, and if so, explain why; (2) If they are indeed robust, clarify how this supports the case for PDM-PURE.
>
>
>
>
> 2. in terms of adaptive attack, the authors said "  The denoisers in PDMs are trained with extensive data augmentation, including Gaussian noise, which makes them quite robust to small perturbations, ...., The input to the network includes a high level of randomness, as new noise is injected at different timesteps, making the whole system includes high level of randomness, making it robust."
>
> If this is the reason the authors believe PDMs are robust against adaptive attacks, I would like to mention a few examples of such attacks. For instance, reference [1] introduces various random augmentations and noise, which cause state-of-the-art attacks to fail against this defense. However, an adaptive attack that simply employs EoT,  with fixed randomness, and does transfer attacks,  can effectively break this defense, as noted in [2]. One more example that breaks diffpure is in [3].
>
> Similarly, what if you were to implement this kind of adaptive attack? Based on experience with attacks and defenses, it seems that only certified robustness and adversarial training provide a guarantee of robustness against adaptive attacks. I would be interested to learn if there is a third approach, such as the method proposed in this paper. If PDM turns out to have some weaknesses against adaptive attacks, what are some valuable lessons we could still take away from that? That's more what I was concerned about.
>
> [1] Ensemble everything everywhere: Multi-scale aggregation for adversarial robustness. arxiv :2408.05446
>
> [2] https://openreview.net/forum?id=IHRQif8VQC. check public comments.
>
> [3] Towards Understanding the Robustness of Diffusion-Based Purification: A Stochastic Perspective.  arxiv  2404.14309

---

> ### Comment · Reviewer_6bSH · 2024-11-29
> **Response to the discussion.**
>
> Dear Reviewer sKMW, thank you very much for your prompt, thoughtful, and detailed response, and for your patience throughout this review process. I am truly impressed by your deep understanding of the field and also your insightful comments and I have learned a lot from your dicussion. I would like to continue our discussion as follows:
>
> - Regarding Paper Writing, I completely agree with you that the manuscript contains numerous typos and grammatical errors. I too have noticed some of these errors in the initial review, and I apologize for any confusion caused. To clarify, my comment was meant to convey that the paper is generally logically coherent and easy to follow, despite these language issues. I fully support your suggestions, and I will encourage the authors to address these concerns thoroughly in the revised manuscript.
>
> - Regarding Adaptive Attacks, I am grateful for your deep understanding of adversarial machine learning and adaptive attacks. I also agree with your point that the term "adaptive attack" is often misused in the current literature. Only defenses with certified guarantees of robustness can genuinely claim resilience to adaptively crafted attacks. Many existing papers tend to label certain heuristic or recent attacks as "adaptive," even when these attacks are not intuitively effective. In my view, these should be regarded as "potential countermeasures" rather than true adaptive attacks. For example, some papers use AutoAttack as an example of an "adaptive attack," even though it often performs worse than PGD.
>
> I agree that the authors' claim that "PDM is robust to adaptive attacks" is somewhat overstated without stronger empirical or theoretical support. As such, I have also recommended that the authors revise this assertion in the manuscript.
>
> That said, I do find the authors' empirical explanation for the robustness of PDM to be reasonable. Since diffusion models are trained under Gaussian noising-denoising, this process shares some similarities with adversarial training, which may contribute to their robustness. Moreover, there is growing evidence in the literature, such as the work by Chen et al. and others exploring diffusion models for adversarial purification, suggesting that diffusion models may offer inherent robustness, at least in classification tasks. So I wonder if this inherent robustness might also apply to generative purification tasks, and I guess the authors' focus is on empirical evidence rather than certified robustness at this stage. As the first paper to investigate the robustness of PDMs under attack within the framework of existing attacks on latent diffusion models, the authors provide valuable empirical results and reasonable explanations. In my humble opinion, this paper somewhat contributes to interested audiences, and so I am inclined to support its acceptance at ICLR.
>
> - Regarding the Ensemble Everything Everywhere Paper: I agree with your assessment of the recent Ensemble Everything Everywhere paper. It is problematic, and as you pointed out, and we cannot learn anything but conducting the right experiments on adaptive attacks is crucial for understanding adversarial robustness. This paper fails to offer significant insights for two reasons: (a) it combines and extends past defense techniques that have been shown to be ineffective, and (b) it does not evaluate its defense against adaptive countermeasures that are known to be potentially effective. For example, the authors do not recognize that their attacks exhibit clear signs of gradient masking, which, while effective against many existing attacks, can be bypassed with simple countermeasures (e.g., the EoT attack as you mentioned). Yet I believe this paper does not satisfy any of these reasons, so I hesitant to agree that breaking it is trivial.
>
> - Regarding the New Adaptive Attack on DiffPure: thank you for pointing out the recent work that challenges DiffPure. I must admit that I have not been closely following the latest developments in this area, as my primary focus lies elsewhere. My previous understanding was that DiffPure offered non-trivial robustness, and I was unaware of this recent work. After carefully reading the paper you referenced, I now realize that it offers important insights. This new attack suggests that DiffPure’s robustness is (mostly) a result of its stochastic nature, and that it does not claim to bypass DiffPure under real black-box setting (the attack is successful under the assumption that the stochasticity is known to the attacker). Therefore, I still believe that diffusion-based purification remains a non-trivial defense to bypass. I would encourage the authors to carefully consider this recent work and include a discussion of it in their revision.
>
> Once again, thank you for your invaluable feedback and constructive suggestions. I believe that these would greatly help the authors.

---

> ### Author Response · Authors · 2024-11-29
> **[Author response] Thanks for the discussion.**
>
> We are pleasantly surprised by the enthusiastic discussion about this work. We deeply appreciate the contributions of Reviewers 6bSH and sKMW, whose insights may helped enhance the clarity and impact of the paper for the broader community.
>
> > About the writing @ Reviewer sKMW
>
> Regarding typos, we did a thorough proof-reading and fixed most of the typos. We believe the paper is easy to read and easy to understand, we will double-check it for the final revision.
>
> Regarding the organization, we kindly disagree with **Reviewer sKMW** that `demonstrating that "attacks for LDM do not work for PDM" and emphasizes the need to focus on PDM.` First we do not think it is intuitive, before this paper, people has a false understanding that DM is vulnerable because LDM is vulnerable, it is worth put strength to show that attacking the diffusion loss will not fool PDMs.
>
>
> > About the adaptive attacks in [3] @ Reviewer 6bSH and sKMW
>
> Here, we would like to elaborate further on [3], as we are very familiar with this line of work.
>
> The original Diff-Pure approach in this context is designed to purify noise so that **classifier** is not misled. To highlight the difference in the pipeline for our Diff-Pure, consider the following:
>
> - **[pipeline 1]** Attacked Image $\rightarrow$ SDEdit of PDM $\rightarrow$ Classifier
> - **[pipeline 2]**  Attacked Image $\rightarrow$ SDEdit of PDM $\rightarrow$ Human-eyes
>
> In both pipelines, we employ the same attacks targeting SDEdit. Reference [a,b] demonstrates that it is possible to attack the entire pipeline to generate samples that can deceive the classifier.
>
> Now, consider the difference in the attack goals:
>
> - To fool Pipeline 1, we only need to deceive the classifier. This is relatively straightforward, as even a small perturbation can suffice to mislead the classifier, and [a] even uses realistic outputs to achieve this.
> - In contrast, to fool Pipeline 2, we must significantly alter the output of the PDM to deceive human observers. This task is considerably more challenging because it requires dramatic changes to the image, which are harder to achieve without being easily detected.
>
> This distinction highlights that the success of Diff-Pure in Pipeline 1 does not necessarily mean something to Pipeline 2, as the robustness criteria differ fundamentally between the two.
>
>
>
>
> [a] Diffusion-Based Adversarial Sample Generation for Improved Stealthiness and Controllability
>
> [b] DiffAttack: Evasion Attacks Against Diffusion-Based Adversarial Purification

---

> > ### Comment · Reviewer_sKMW · 2024-12-01
> > **reply**
> >
> > Thanks for the further discussions!
> >
> > > In contrast, to fool Pipeline 2, we must significantly alter the output of the PDM to deceive human observers. This task is considerably more challenging
> >
> > One important point to highlight is that adversarial examples capable of fooling human eyes do not necessarily imply adversarial robustness. Additionally, such examples do not always require dramatic changes to the image.
> > For example, evidence presented in [1] demonstrates that while this method can produce adversarial examples detectable by human observers, these examples are neither robust nor involve more than small perturbations.
> >
> > Given that current attacks already provide some insights into how to target PDM[2,3], I suggest that the authors conduct a more rigorous evaluation first (This is particularly important, as your follow-up method heavily relies on this). From my understanding, it is not entirely accurate to claim that "PDMs are robust against adaptive attacks," as there are potential attack strategies, some of which we discussed earlier.
> >
> >
> >
> >
> > [1]  Ensemble everything everywhere: Multi-scale aggregation for adversarial robustness. arxiv :2408.05446
> >
> > [2] Towards Understanding the Robustness of Diffusion-Based Purification: A Stochastic Perspective. arxiv 2404.14309
> >
> > [3] DiffAttack: Evasion Attacks Against Diffusion-Based Adversarial Purification

---

> > > ### Author Response · Authors · 2024-12-02
> > > **reply**
> > >
> > > > One important point to highlight is that adversarial examples capable of fooling human eyes do not necessarily imply adversarial robustness. Additionally, such examples do not always require dramatic changes to the image.
> > >
> > >
> > > Here we would like to kindly remind the reviewer about the settings of this paper, as our title `Rethinking Adversarial Attacks as Protection Against Diffusion-based Mimicry` said, we focus on adversarial perturbation as protection for diffusion-based mimicry. The threat model is clearly defined for this literature and we kindly remind that `adversarial examples capable of fooling human eyes do not necessarily imply adversarial robustness` because it is far away from the task and threat models for this paper.
> > >
> > > > Given that current attacks already provide some insights into how to target PDM[2,3], I suggest that the authors conduct a more rigorous evaluation first (This is particularly important, as your follow-up method heavily relies on this).
> > >
> > >
> > >
> > > [2,3] focus on attacking PDM as a purifier for classifiers; however, their settings differ significantly from those in our paper. Since classifiers are inherently vulnerable, basing results on attacks targeting the classifier to claim that PDM is vulnerable is not particularly meaningful.
> > >
> > > You can imagine if you replace PDM with a Gaussian blur, it is still easy to attack the whole pipeline, but you cannot say Gaussian blur is vulnerable, it can always generate meaningful images which is realistic.
> > >
> > > Our contributions are acknowledge by other reviewers and we do think this paper provides critical insights for the community.

---

### Official Review · Reviewer_xkto · 2024-11-03

**Soundness:** 3
**Presentation:** 2
**Contribution:** 2
**Rating:** 6
**Confidence:** 4

**Summary:**

This paper raises two concerns against IP protection methods based on adversarial attacks to diffusion models, both based on the observation that PDMs (pixel space diffusion models) are more robust to adversarial attacks than LDMs (latent diffusion models).

concern 1. The adversarial attack-based IP protection methods are less effective for PDMs.

concern 2. Using PDM-based purification to preprocess "protected" images can also reduce the effectiveness of adversarial attack-based IP protection methods.

These two are the primary contributions of the submission.

**Strengths:**

The limitations of existing protection methods are indeed something worth highlighting, and the paper did a fair job in providing experimental supports to primary claims.

**Weaknesses:**

There are a few weaknesses but I think they should be fixable:
1. Inaccurate position of their contributions among existing research:
Adversarial robustness of diffusion models (or PDMs) and using diffusion purifications to mitigate adversarial attacks have already been investigated in existing research. While the authors have already cited DiffPure (Nie et al., 2022) in the submission, the usage of DiffPure-like techniques and their contributions are without a doubt outlooked in the story told. This should be fixed by acknowledging (in a much more prominent way) the contributions of existing work adversarial robustness of diffusion models (or PDMs) and using diffusion purifications to mitigate adversarial/other noises.

2. The quantitative results for the PDM-Pure part seem incomplete in a sense that only one metric, FID-score, was reported.
Since 3 other metrics were reported in your Quantitative Measurement of PGD-based Adv-Attacks for LDMs and PDMs (table 2), I believe the authors have no trouble finding more quantitative metrics for evaluating PDM-Pure. I would suggest authors to incorporate these as well.

**Questions:**

Please see weakness for my concerns, where I have included details already.

**Details Of Ethics Concerns:**

There are potentially copyrighted images in the paper, which may or may not be subject to copyright issues.

---

> ### Author Response · Authors · 2024-11-17
> **Thanks for your careful review**
>
> We express our gratitude to the reviewer for conducting a careful review and providing valuable feedback. Here I will answer some questions that can hopefully address your concern:
>
> > W1: Comparison with Diff-Pure (Nie et al., 2022) and other similar works.
>
> That's a great point. While PDM-Pure resembles Diff-Pure in structure, it addresses a different scenario. In most research, such as Diff-Pure, diffusion models with SDEdit are employed to mitigate adversarial perturbations in classification tasks. Our focus, however, is on removing adversarial noise that targets diffusion models directly.
>
> Both PDM-Pure and Diff-Pure operate on the assumption that diffusion models can eliminate out-of-distribution noise patterns through a diffuse-then-denoise process. However, our motivation diverges: given the insight that PDMs are exceptionally robust to adversarial attacks, we hypothesize that they can mitigate "any" noise using the Diff-Pure pipeline, provided it’s sufficiently robust. LDMs, by contrast, cannot achieve this due to their vulnerability to adversarially crafted noise. Notably, Diff-Pure did not highlight the adversarial robustness of PDMs, which is a core focus of our paper.
>
>
> > W2: Other metrics for PDM-Pure experiments
>
> Thanks for pointing it out. We add additional results about other metrics in the Appendix, please refer to ** Revision Summary ** for the updates.

---

> > ### Comment · Reviewer_xkto · 2024-11-20
> >
> > Thanks for the rebuttal.
> >
> > For W2, I consider the addition of metrics positive and encourage the authors to add other metrics as well for future versions. I don't think this will be a big issue during the review process.
> >
> > However, I am not really satisfied with the authors' response regarding W1. I believed the authors may have misinterpreted the main point of W1. To clarify, the issue was NOT that "because there are previous work with similar insights, the contribution of this paper was not enough"; The issue is, this paper does not properly position its contribution among the relevant work, with the ones cited but excluded in the story told. And my suggestion was, and I quote my original review, "This should be fixed by acknowledging (in a much more prominent way) the contributions of existing work adversarial robustness of diffusion models (or PDMs) and using diffusion purifications to mitigate adversarial/other noises."
> >
> > This is still a fixable issue and I positively assume that authors will be able to & willing to fix this. Thus I am keeping my score for now and only slightly leaning towards acceptance, despite that my primary concern has not been addressed yet.

---

> > > ### Author Response · Authors · 2024-11-20
> > > **Thanks for your response.**
> > >
> > > Thanks for your response and again for your careful review.
> > >
> > > > W1
> > >
> > > Thank you for pointing that out. We are willing the fix this issue. In the next revision (maybe in two days after we combining the new responses from all the reviewers), we will include more detailed discussion of Diff-Pure in the section of PDM-Pure.
> > >
> > >  However, regarding the adversarial robustness of PDM, our work is indeed the first to explore this area, particularly in using adversarial noise to safeguard images from diffusion models.

---

> > > > ### Author Response · Authors · 2024-12-03
> > > > **reply**
> > > >
> > > > We apologize for forgetting to include it in the revision v2. To address your remaining concern, we will add the following paragraph in section 5:
> > > >
> > > > `
> > > > Existing works, such as Nie et al. (2022), leverage diffusion models to purify adversarial noise targeting deep classifiers. The adversarial noise in their scenario is specifically crafted to deceive deep classifiers. They employ SDEdit with diffusion models, operating under the assumption that the diffusion process inherently diffuses the adversarial noise as well.
> > > > In contrast, our setting deals with adversarial noise designed to fool diffusion models themselves. We rely on PDMs (Probabilistic Diffusion Models) due to strong empirical evidence supporting their robustness against adversarial noise.
> > > > `

---

### Author Response · Authors · 2024-11-17
**Revision Summary v2**

We made slight revision to the original paper, the details are as follows:

- We do a proof-reading of the paper

Thanks for the careful review of all reviewers (especially sKMW and 6bSH), we have fixed all the typos mentioned in this paper.

- More results of metrics for Table 2:

We add IA-Score as metric into Table2, the results are put in **Appendix Table 3**.

- More results of adaptive attacks for PDMs (To Reviewer sKMW, 6bSH):

We put more results of adaptive attacks for PDMs in **Appendix Figure 17**. We put results of attacks including applying EoT and latent attacks of U-Net middle block.

- Results of using LDM as purifier (To Reviewer sKMW):

We show that using LDM + DiffPure will not be able to purify the protective noise in **Appendix Figure 18**.  We generate protective noise using AdvDM and use SDXL as the purifier; the SDXL will be easily fooled when we apply DiffPure with it.

---

### Author Response · Authors · 2024-12-04
**Summary of Rebuttal**

Dear AC & Reveiwers,

We appreciate the effort and insights provided by all of you, which have been instrumental in improving our work. While some concerns about adaptive attacks and the robustness claims remain under discussion, we believe we have addressed the majority of the key points raised. Below, we summarize how the reviewers’ feedback has been incorporated:

**Novelty and Contribution:**

Most reviewers (xkto, GDQ5, 6bSH) acknowledged that our work addresses a critical gap in understanding the robustness of PDMs against adversarial attacks, contrasting with the widely studied LDMs. It is an important insights for using adversarial perturbation to protect our personal images.
We have clarified the positioning of our contributions relative to previous works like DiffPure, emphasizing that our focus on the inherent adversarial robustness of PDMs is novel.


**Empirical Results:**

We extended the experimental evaluation by including additional metrics and ablation studies, as suggested by the reviewer xkto.
To address concerns about adaptive attacks, we discussed the limitations of existing attacks and proposed plausible directions for future research.


**Presentation and Clarity:**

We thoroughly proofread the manuscript, addressing typos and improving clarity.


**Discussion with Reviewer sKMW and Reviewer 6bSH**

Our discussions with reviewer sKMW were particularly rigorous, focusing on their concerns regarding (1) the manuscript's writing quality, (2) the treatment of adaptive attacks, and (3) comparisons with existing work targeting DiffPure. We have addressed each of these points comprehensively in the discussion section.

We were encouraged by reviewer 6bSH, who participated in these discussions and expressed support for our work. Their positive remarks re-affirm our belief that our contributions are valuable and that our findings hold critical relevance for the community.


Thank you for your time and efforts.

Sincerely,

The Authors of Submission 8689

---

### Meta-Review · Area_Chair_yabs · 2024-12-22

**Metareview:**

This paper presents an interesting finding that pixel space diffusion models (PDMs) are more robust against several existing attacks compared to latent diffusion models (LDMs). Most prior work on attacking diffusion models was based on LDMs due to their popularity, and the research on attacking PDMs is limited in prior work. The paper is overall written very well and the story is attractive, so two reviewers support the acceptance of the paper. However, the AC shares the same concern as Reviewer sKMW that the conclusion may not be fully supported because of the lack of specialized attacks on PDMs. Although the paper claimed to conduct "adaptive" attacks on PDMs, the methodology used, such as targeted/untargeted loss and EOT, were not specially designed to attack PDMs, and it is expected that they do not work well here - more efforts are need before claiming a model is unbreakable. In addition, the claim that the weakness of LDMs is from the encoder needs more justification and support, both theoretically and experimentally. Since none of the claims in the paper were justified in theory, the experiment section (pages 8 - 9) should be expanded to include more observations and insights. Considering the novel insights in this paper and these weaknesses, the AC believes this is a borderline paper and can benefit from more work to make the claims more convincing.

**Additional Comments On Reviewer Discussion:**

Reviewers had great discussions and they believe the findings in the paper are interesting and the story is good. One key issue in the discussion is whether the paper considered sufficiently strong attacks on PDMs, and reviewers 6bSH and sKMW had a debate regarding this matter. After carefully checking the paper and the discussions among the reviewers and authors, the AC's opinion about the paper is that this is a borderline paper that can benefit from more support for its claims, so I inclined to reject the current version of this paper.

---

### Decision · Program_Chairs · 2025-01-22

Reject